# communications
## engineering

# Drone swarm strategy for the detection and tracking of occluded targets in complex environments

Rakesh John Amala Arokia Nathan[1], Indrajit Kurmi[1] & Oliver Bimber [1✉]

Drone swarms can achieve tasks via collaboration that are impossible for single drones alone. Synthetic aperture (SA) sensing is a signal processing technique that takes measurements from limited size sensors and computationally combines the data to mimic sensor apertures of much greater widths. Here we use SA sensing and propose an adaptive real-time particle swarm optimization (PSO) strategy for autonomous drone swarms to detect and track occluded targets in densely forested areas. Simulation results show that our approach achieved a maximum target visibility of 72% within 14 seconds. In comparison, blind sampling strategies resulted in only 51% visibility after 75 seconds and 19% visibility in 3 seconds for sequential brute force sampling and parallel sampling respectively. Our approach provides fast and reliable detection of occluded targets, and demonstrates the feasibility and efficiency of using swarm drones for search and rescue in areas that are not easily accessed by humans, such as forests and disaster sites.

[1] Computer Science Department, Johannes Kepler University Linz, Linz 4040, Austria. ✉email: oliver.bimber@jku.at

Drone swarms often explore a collaborative behavior to perform as an intelligent group of individuals and achieve objectives that would be impossible or impractical to achieve individually[1–5]. Single drones within the swarm can perceive their local environment and then act accordingly with or without direct awareness of the swarm's overall objective. They have been used for surveillance and environment mapping[6–10], airbase communication networking[11–13], target detection and tracking[14–17], infrastructure inspection and construction[18–20], or load transport and delivery[21], and are particularly useful in areas that are not easily accessible by humans, such as forests and disaster sites[22–24].

Drone swarms utilize either a centralized control system[6,7,13] (where either an omniscient drone or an external computer performs communication and pre-plans actions) or a decentralized control system[8,10,11,20,24] (drones communicate with each other and make decisions locally). Hand-crafted[1,25,26] or automatically designed algorithms (e.g., heuristics-based, evolutionary-based, learning-based[1,2,27–37]) have been applied to implement various swarm behaviors (e.g., flocking, formation flight and distributed sensing) for achieving a desired objective (e.g., path planning, task assignment, flight control, formation reconfiguration and collision avoidance).

Meta-heuristic algorithms (e.g., genetic algorithms, differential-equation-based algorithms, ant-colony and particle swarm optimization (PSO)) exploit non-Markovian properties of a problem (i.e., each individual drone has only a partial observation of the swarm). Their inherent ability to deal with credit assignment has led to these approaches being more widely adopted than classical and learning-based algorithms (e.g., deep-learning neural network, reinforcement-learning) in the swarm literature. PSO in particular is computationally efficient, robust, and can be incorporated into hierarchical planning structures.

Synthetic aperture (SA) sensing is a signal processing technique that takes measurements of limited size sensors and improves them by computationally combining their samples to mimic sensor apertures of physically impossible width. In recent decades, it has been applied in various fields, such as radar[38–40], radio telescopes[41,42], interferometric microscopy[43], sonar[44,45], ultrasound[46,47], LiDAR[48,49], and imaging[50–52].

With airborne optical sectioning (AOS)[53–64], we have introduced an optical SA imaging technique for removing partial occlusion caused by vegetation (cf. Fig. 1a). Drones equipped with conventional cameras are used to sample images above forest. These images have a wide depth of field, due to the cameras' narrow apertures (i.e., small lenses, usually a few millimeters). They are registered and integrated to mimic shallow depth of field images as would have been captured by a very wide-aperture camera (using a lens that covers the sampling area, several meters in diameter). Computationally focusing the resulting integral image on the forest ground by registering the single images appropriately with respect to the drones' sampling positions emphasizes the targets' signal while very quickly suppressing (due to the shallow depth of field) the signals of occluders above. The unique advantages of AOS, such as its real-time processing capability and wavelength independence, open many new application possibilities in contexts where occlusion is problematic. These include, for instance, search and rescue, wildlife observation, wildfire detection, and surveillance. We have demonstrated that image processing techniques, such as classification[61] and anomaly detection[63], are significantly more efficient in the presence of occlusion when applied to integral images rather than to single images.

We have previously presented several sampling techniques for AOS. Early approaches used single, sequentially sampling drones that followed predefined waypoints[53–61] or autonomously determined flight paths that were dynamically planned based on classification results[62] (cf. Fig. 1b). Sequential sampling does not support moving targets, as long sampling periods result in strong motion blur. Initially, parallel sampling strategies were investigated, which used large 1D camera arrays with a fixed sampling pattern instead of single cameras (cf. Fig. 1c). They supported motion detection and tracking, but were cumbersome to handle, difficult to fly stably, and resulted in undersampled integrals. In all of these cases, varying forest properties, such as changing local occlusion densities, could not be considered for optimized sampling. Due to their high complexity, local viewing conditions could not be reconstructed in real time during scanning, and were impossible to model or learn because of their high degree of randomness. However, knowing sparser forest patches through which targets could be observed with less occlusion and possibly even from more oblique viewing angles could make AOS sampling significantly more efficient than sampling blindly. The sampling pattern could then adapt adequately and dynamically to local viewing conditions.

In this article, we demonstrate that PSO is a suitable instrument for solving this problem. Here, we consider an autonomous swarm of drones that explores optimal local viewing conditions for AOS sampling. They approximate the optical signal of extremely wide and adaptable airborne lens apertures (see Supplementary Movie Abstract).

While PSO has a long history in modeling real-time swarm behavior[32–37], the objective function for AOS is not constant (especially in the case of target motion), highly random (forest occlusion), neither linear nor smooth, and certainly not differentiable. We present a PSO variant and a new objective function that can cope with these challenges and that considers additional sampling constraints, such as enforcing minimal sampling steps because smaller ones would not contribute to occlusion removal[54,57]. Furthermore, we explain how PSO hyperparameters are directly related to SA properties, such as the aperture diameter, and show that collision avoidance can be achieved by simple altitude offsets without significant reduction in sampling quality.

Experiments reveal that, in contrast to previous blind sampling strategies, which are based on predefined waypoint flights (where sampling is either sequential or parallel), drone swarms adapt to optimal viewing conditions, such as low occlusion density and large target view obliqueness (which increases the projected footprint of the target). Furthermore, they combine sequentially and parallelly recorded samples. Both increase visibility significantly while reducing sampling time. Lastly, moving targets can be detected and tracked even under difficult through foliage conditions.

We acknowledge that blind brute force sampling with an infinite number of samples over the entire search area may yield similar visibility results, but is unsuitable for time-sensitive applications such as search and rescue operations where time and drone battery life are limited. To address this limitation, the proposed adaptive swarm sampling approach is evaluated against blind sampling methods (sequential and parallel) based on visibility (%) and time (seconds), demonstrating superior performance. This comparison is considered equitable because the primary goal is to locate the target swiftly and reliably, which is an optimization issue that cannot be resolved by dense brute force search due to performance and complexity restrictions.

Our approach using autonomously exploring swarms of drones can lead to faster and much more reliable detection of strongly occluded targets, such as missing persons in search and rescue missions, animals during wildlife observation, hot spots during wildfire inspections, or security threats during patrols.

## Results

We applied a procedural forest model (cf. Fig. 2a) to simulate the AOS sampling process in four spectral bands (visible and far infrared, RGB+thermal), for different forest occlusion densities and sampling procedures (parallel, sequential, single drones, camera arrays, swarms), as well as for moving and static targets. Implementation details are summarized in Implementation, and a comparison between simulated and real integral images is

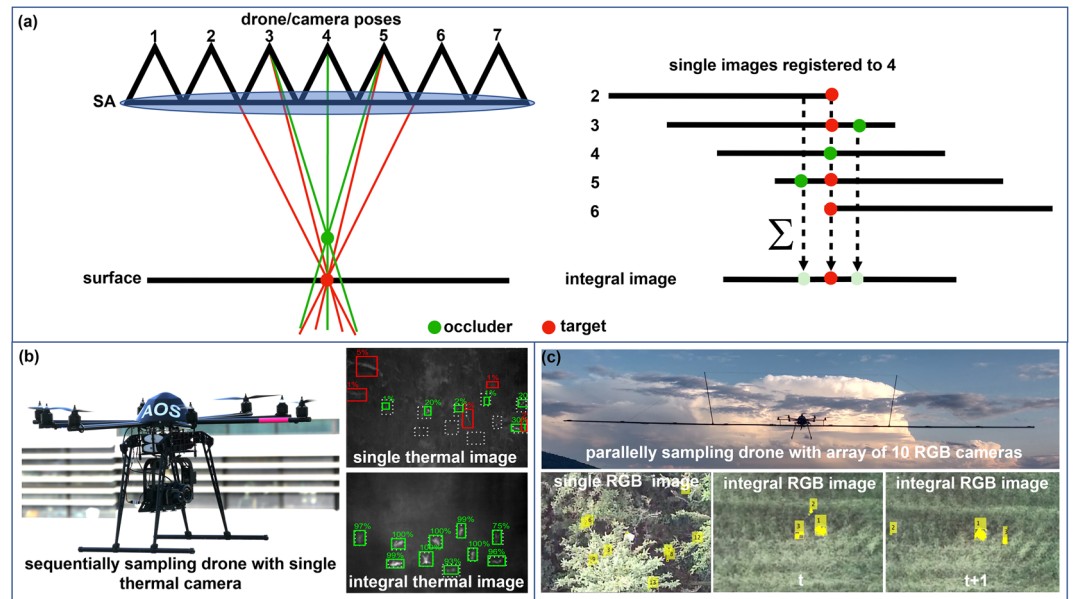

**Fig. 1 Blind sampling strategies for target detection and tracking in forested environments using airborne optical sectioning. a** Airborne optical sectioning sampling principle: after image registration and integration, misaligned occluders above the focused ground surface are suppressed, while aligned targets on the ground surface are emphasized. **b** A single-camera drone prototype that sequentially samples the synthetic aperture (SA) to detect static targets (persons lying on the ground) in a dense forest[61, 62]. **c** A drone prototype with a parallelly sampling camera array that spans a 10-m wide 1D SA supports the detection and tracking of moving targets (walking persons) in a dense forest[63]. **b** Classification as well as **c** anomaly detection and tracking through foliage becomes possible in integral images across multiple time steps *t* while it remains unfeasible in single images (with many false positives and true negatives).

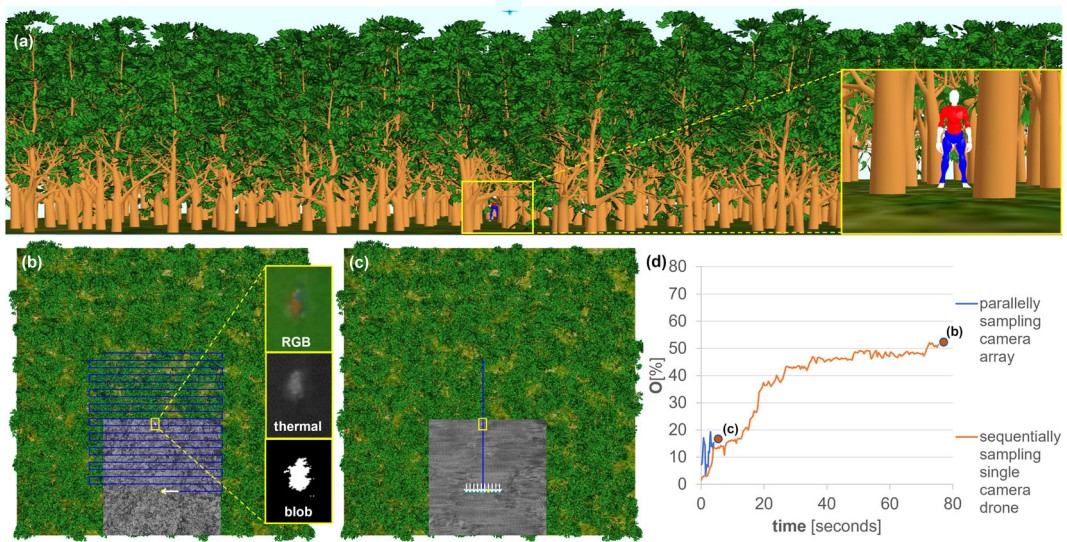

**Fig. 2 Impact of blind sampling strategies on target visibility in a simulated procedural forest environment. a** Our simulation environment was a 1 ha procedural forest with one hidden avatar. **b** Blind brute force sequential sampling, as in the case of a single-camera drone that sequentially samples the SA (Fig. 1b)[61], **d** led to a maximum target visibility (MTV) of 51% after a long period of 75 s. **c** Blind parallel sampling, as with an airborne camera array (Fig. 1c)[63], was fast, but **d** resulted in only 19% MTV after a short period of 3 s. Our objective function *O* models target visibility by the contour size of the largest connected pixel cluster (blob) computed from anomalies in color (RGB) and thermal channels, as explained in Objective Function. Note that the yellow boxes highlight the target, the white arrows show the movement of the drones between time steps *t* − 1 and *t*, the blue lines represent the total sampling paths, and the gray area shows the integrated ground coverage at time *t*. Simulation parameters: drone's ground speed = 10 m/s, forest density = 300 trees/ha, **b** single-camera drone sampling sequentially a 36 × 38 m SA with a 4 × 2 m resolution, **c** array of 10 cameras sampling at 1 m inter-camera distance with 2 m steps in the flight direction (as shown for the prototype in Fig. 1c). See Supplementary Movie 1.

provided in Supplementary Note 1 (cf. Supplementary Fig. S1). All experiments were evaluated with the objective function presented in Objective function. It determines the target visibility, and we consider it in % (given that the highest visibility of an unoccluded target is known and 100%). Note that the maximum target visibility (MTV) under occlusion over the entire sampling time provides the best hint of a potentially detected target. Our PSO approach, with its collision avoidance strategy and hyper-parameters, is explained in Particle swarm optimization, Collision avoidance, and Hyper-parameters. We conclude with a discussion of the results in the Discussion and conclusion.

The goal of the following experiments was the automatic detection of a standing or walking person in occluding forest. We compare different sampling options for this task: (1) a single sequentially sampling drone (as shown in Fig. 1a) following a predefined 2D search grid at constant altitude (40 m AGL), (2) an airborne array of 10 parallelly recording cameras (as shown in Fig. 1b) while following a predefined 1D search path at constant altitude (40 m AGL), and (3) swarms of 3, 5, and 10 drones following our PSO at various individual altitudes for collision avoidance (swarm average altitude is always 40 m AGL), as explained in Collision avoidance. In all cases, starting conditions and flight speed (10 m/s) were identical, and the images captured (sequentially and parallelly) were integrated in overlapping ground surface regions. We also investigated different forest densities: sparse (300 trees/ha), medium (400 trees/ha), and dense (500 trees/ha).

Previous blind strategies that sample based on predefined waypoints (i.e., sequentially with a single-camera drone or parallelly with camera arrays) were reconsidered in the experiment shown in Fig. 2a–d. The single-camera drone that carries out blind brute force sampling has a SA size similar to that of the adaptive swarm approach in the default scanning pattern. In contrast, the blind parallel sampling camera array has a SA size of only 9 m. Larger arrays are impractical because they are difficult to maintain and do not allow proper flight stabilization due to the extreme lever arms, as explained in ref. [63]. While brute force sequential sampling led to a relatively high visibility improvement through occlusion removal after a relatively long time (51% MTV after 75 s), parallel sampling resulted very quickly with only marginal visibility improvements (19% MTV after 3 s). In both cases, local occlusion densities of the forest or view obliqueness of the target were not considered. Note that the geometric distribution behavior of visibility improvement that can be observed for sequential sampling with an increasing number of integrated images matches the findings made with the statistical model described in ref. [54].

Adaptively sampling drone swarms that consider local occlusion density and target view obliqueness, however, can significantly increase visibility while reducing sampling time, as shown in the next experiment (Fig. 3a–e). Under the same conditions as for the experiments in Fig. 2, an MTV of 72% was reached after 14 s because the swarm found and converged over gaps or sparse density regions in the vegetation while preferring oblique target views. Our PSO that guides the swarm behavior considers sequentially as well as parallelly captured samples for maximizing target visibility, as explained in detail in Methods.

The size of the swarm clearly matters, as shown in the experiment in Fig. 4a–c, f. Larger swarms profit from a wider SA and a denser sampling. They consequently led to better visibility (max. 72% for $n = 10$, 35% for $n = 5$, and 22% for $n = 3$) and larger sampling coverage. However, if the swarm becomes too large or the SA is extremely wide, then the field of view coverage of the drones' cameras (i.e., the overlapping ground region covered by the swarm) is reduced to zero for drones at the SA's periphery.

With increasing forest density, visibility decreases because of denser occlusion, as can be seen in the experiment in Fig. 4c–e, g. By repeating the experiment shown in Fig. 3, MTV drops from 72% (300 trees/ha) down to 42% (400 trees/ha) and 31% (500 trees/ha). The averaged outcomes of three simulation runs for each of the six scenarios illustrated in Fig. 4 are detailed in Supplementary Note 3 (cf. Supplementary Fig. S3). Note that our simulations were conducted for an extended period of time but the plots in Figs. 2–4, were cut off at the point where there was no further improvement observed in the objective. The full simulation results are available in the Supplementary material.

Moving targets can also be detected and tracked. For the experiment shown in Fig. 5a–i, the average differences between the target's ground truth position, motion speed, motion direction and the corresponding estimations of our PSO were 0.59 m, 0.09 m/s, and 9.21/circ, respectively. When the target leaves the swarm's view, the swarm starts to diverge into the default scanning pattern toward the last known target position. When the target stops, the swarm mainly converges into a circular SA pattern. If the target is inside the swarm's view while moving, the swarm converges and diverges depending on the local occlusion situation. Note that its convergence and divergence behavior may also be attributed to the target moving in and out of the swarm's field of view. This is illustrated in Fig. 5i. Supplementary Note 4 presents a further motion example (cf. Supplementary Fig. S4), while Supplementary Notes 5 and 6 show failure cases for too fast target motion (cf. Supplementary Fig. S5) and locally too dense occlusion (cf. Supplementary Fig. S6).

A detailed discussion of these results is provided in the Discussion and conclusion.

## Methods

**Particle swarm optimization.** In order to determine the particle positions at time $t + 1$, classical PSO algorithms[65–68] add the following velocity vectors to the current positions at time instance $t$:

$$V_i^{t+1} = c_0 \cdot (V_i^t) + c_1 \cdot r_1 (P_{\text{best}}^i - P_i^t) + c_2 \cdot r_2 (G_{best} - P_i^t), \tag{1}$$

where $V_i^t$ is the velocity vector of particle $i$ at time $t$, $P_{\text{best}}^i$ is the position of best objective ever explored by particle $i$, $G_{\text{best}}$ is the position of the best objective ever explored by any particle, $r_1$ and $r_2$ are random numbers (0…1), and $c_0$, $c_1$, $c_2$ are the hyper-parameters. Here, $c_0$ is the inertia weight constant (i.e., how much of the previous velocity is preserved), $c_1$ is the cognitive coefficient, which refines the results of each particle, and $c_2$ is the social coefficient, which refines the results of the entire swarm.

For our problem, two main observations can be made:

(1) Our objective function is based on randomness (forest occlusion) and is (especially in the case of target motion) not constant. It is neither linear nor smooth, and certainly not differentiable. The latter is the reason for choosing PSO in general, as gradient-decent-based optimizations are not possible.

(2) A bias toward the positions with the best sample values over history (either for a single particle $P_{\text{best}}^i$ or for the global swarm $G_{best}$) is not effective if dynamics and randomness affect the objective function. Sampling multiple times at the same (best) position does not improve the objective in our case, as no additional unoccluded parts of the target can possibly be seen. Therefore, particles must always remain dynamic in our case (and not converge to a single position), and must cover the SA effectively.

To address these two observations, our PSO approach must be much more explorative than exploitative:

(1) The swarm behavior itself is constrained to the current time instance (i.e., not considering history): random local explorations

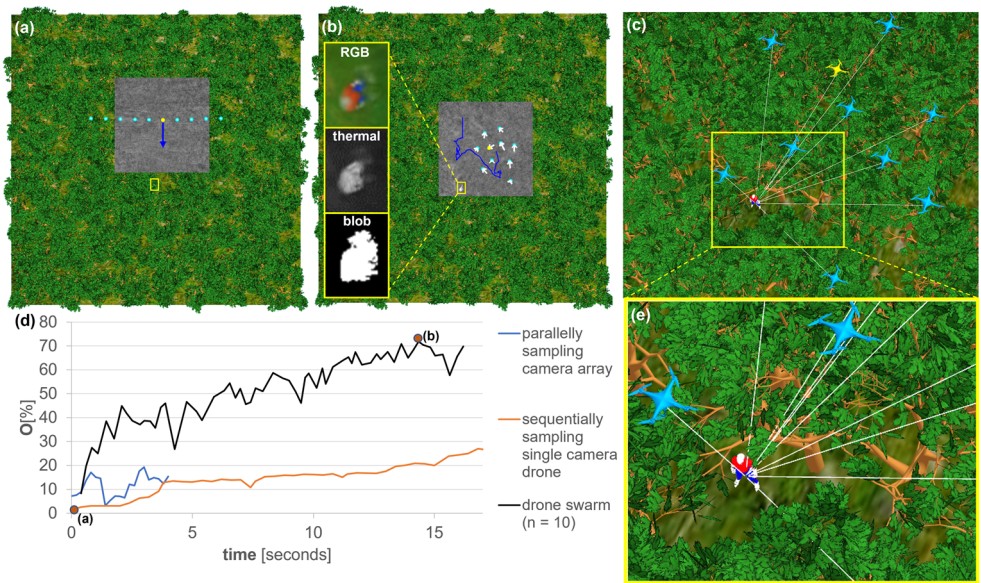

**Fig. 3 Particle swarm optimization for enhanced target visibility. a** Swarm of 10 drones approaches target in default (linear) scanning pattern and **b** then converges **d** to maximize target visibility. **b** Our objective function $O$ models target visibility by the contour size of the largest connected pixel cluster (blob) computed from anomalies in color (RGB) and thermal channels. **d** A maximum target visibility (MTV) of 72% was reached after 14 s by finding and converging above gaps in the vegetation, **c**, **e** as in the close-ups. For comparison, the blind sampling results from Fig. 2 are also **d** plotted for the same duration. Note that the white rays (solely used for visualization purposes) indicate direct line of sight between drones and target, the yellow boxes highlight the target, the white arrows show the movement of the drones between time steps $t-1$ and $t$, the blue lines represent the total sampling paths of the swarm's center of gravity, the yellow dots/drones indicate the best sampling position at time $t$, and the gray area shows the integrated ground coverage at time $t$. Simulation parameters (see Methods for details): drones' ground speed = 10 m/s, forest density = 300 trees/ha, $n = 10$, $T = 16.3\%$, $\Delta_h = 1\,\mathrm{m}$, $h_1 = 35\,\mathrm{m}$, $c_1 = 1\,\mathrm{m}$, $c_2 = 2\,\mathrm{m}$, $c_3 = 2\,\mathrm{m}$, $s = c_4 = 4.2\,\mathrm{m}$, $c_5 = 0.3$. See Supplementary Movie 2.

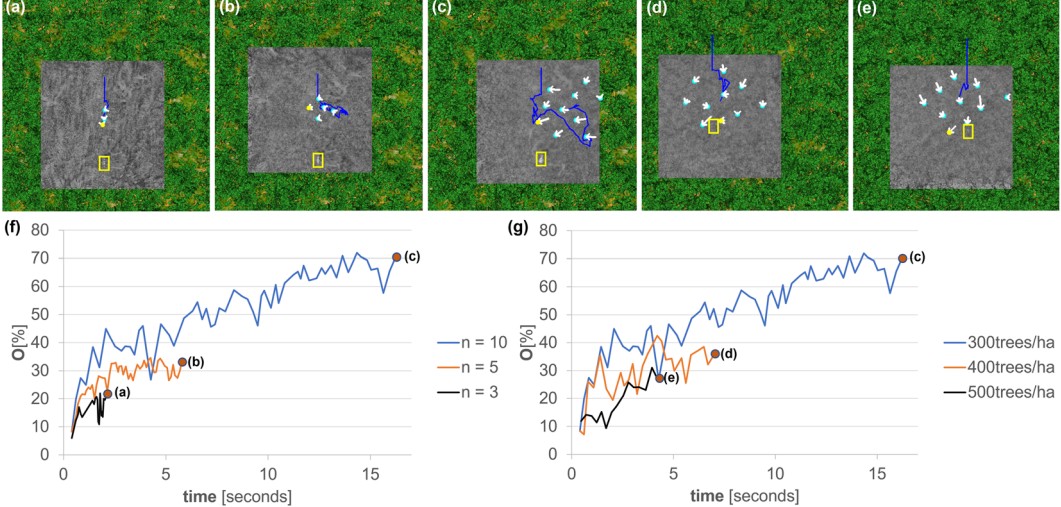

**Fig. 4 Effects of varying swarm size and forest density on target visibility.** Increasing swarm size ($n = 3, 5, 10$ in **a–c**) **f** led to better target visibility. Here, a wider synthetic aperture and a larger number of samples increased the target visibility and consequently the probability of its detection. Increasing forest density (300, 400, 500 trees/ha in **c–e**) **g** decreased target visibility due to denser occlusion. Here, the swarm size was constant ($n = 10$, as in Fig. 3). Note that the yellow boxes highlight the target, the white arrows show the movement of the drones between time steps $t-1$ and $t$, the blue lines represent the total sampling paths of the swarm's center of gravity, the yellow dots indicate the best sampling position at time $t$, and the gray area shows the integrated ground coverage at time $t$. Note also that the plots end in case of no further visibility improvement. Simulation parameters (see Methods for details): drones' ground speed = 10 m/s, $\Delta_h = 1\,\mathrm{m}$, forest density = **a–c** 300 trees/ha, **d** 400 trees/ha, **e** 500 trees/ha, $n =$ **a** 3, **b** 5, **c–e** 10, $T =$ **a** 7.93%, **b** 11.19%, **c** 16.3%, **d** 8.86%, **e** 8.39%, $h_1 =$ **a** 38 m, **b** 39 m, **c–e** 35 m, $c_1 =$ **a** 0.22 m, **b** 0.445 m, **c** 1 m, **d** 1 m, **e** 1 m, $c_2 =$ **e** 0.44 m, **b** 0.89 m, **c** 2 m, **d** 2 m, **e** 2 m, $c_3 =$ **a–e** 2 m, $s = c_4 =$ **a** 0.933 m, **b** 1.87 m, **c** 4.2 m, **d** 4.2 m, **e** 4.2 m, $c_5 =$ **a–e** 0.3. See Supplementary Movie 3.

of particles with a bias toward a temporal global leader (i.e., the best sample at the current time instance), enforcing a minimal distance constraint that defines the SA properties.

(2) The objective function is conditionally integrated (i.e., a new sample is integrated only if it improves the objective), using

parallel samples (i.e., samples taken at the current time instance) and sequential samples (i.e., samples taken at previous time instances).

(3) Instead of an inertia weight constant, we apply a condition (i.e., if nothing is found) bias toward a default scanning pattern.

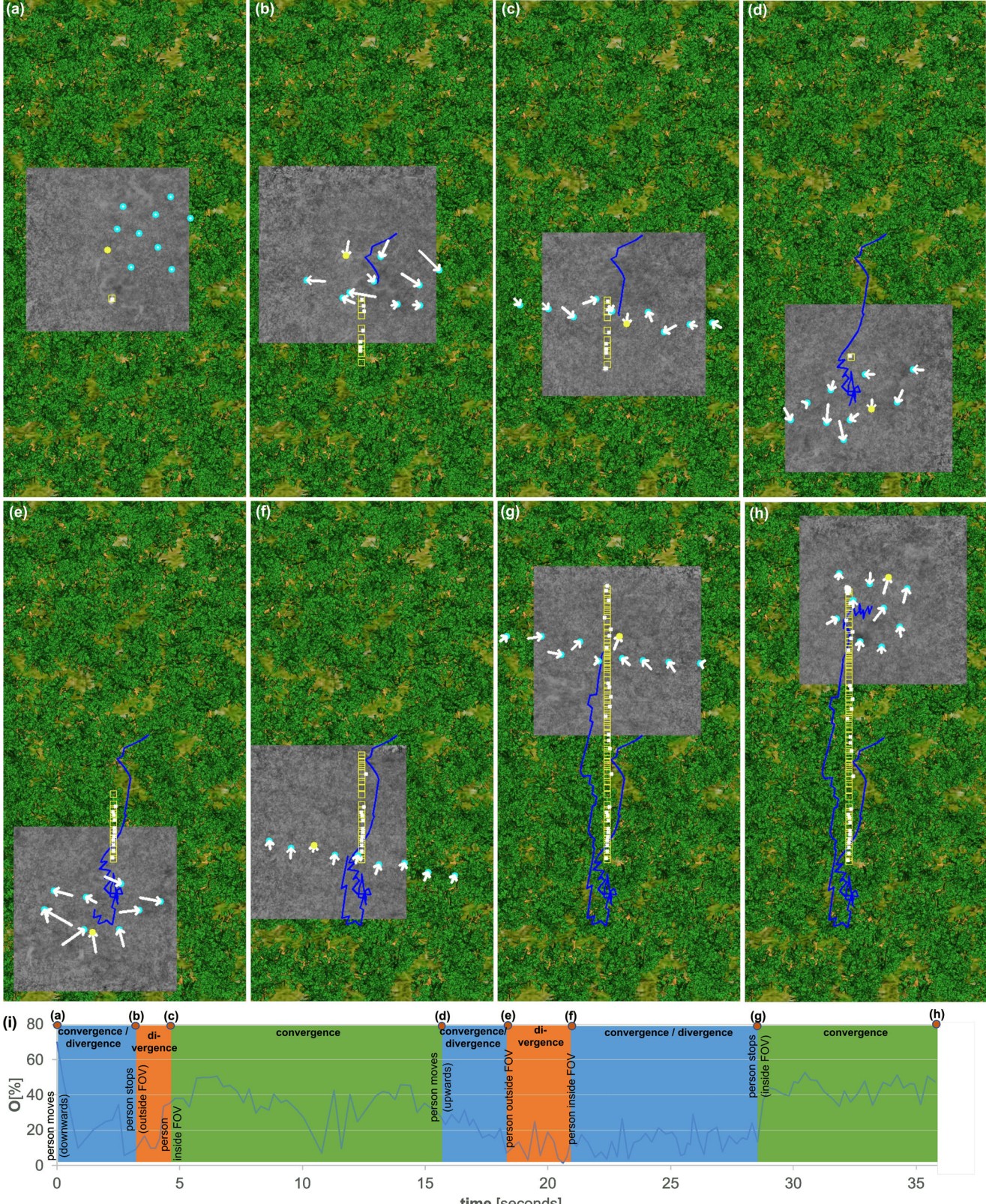

**Fig. 5 Detection and tracking of moving target. a** Starting from the final state shown in Fig. 3, a walking person (speed: 4 m/s) was simulated: **a**, **b** moving 13 m downwards, **b**–**d** resting for 12 s, **d**–**g** moving 55 m upwards, **g**, **h** resting. **i** The target visibility while the swarm is tracking the person is shown, and the different convergence and divergence phases of the swarm are highlighted. Note that the yellow boxes highlight the target's ground truth positions, the white stars indicate the target's estimated positions, the white arrows show the movement of the drones between time steps $t-1$ and $t$, the blue lines represent the total sampling paths of the swarm's center of gravity, the yellow dots indicate the best sampling position at time $t$, and the gray area shows the integrated ground coverage at time $t$. Simulation parameters (see Methods for details): drones' ground speed $= 10$ m/s, $\Delta_h = 1$ m, forest density $= 300$ trees/ha, $n = 10$, $T = 16.3\%$, $h_1 = 35$ m, $c_1 = 1$ m, $c_2 = 2$ m, $c_3 = 1.643$ m, $s = c_4 = 4.2$ m, $c_5 = 0.3$. See Supplementary Movie 4.

Algorithm 1 summarizes one iteration of our PSO approach at time instance $t$.

**Algorithm 1.** PSO Iteration

**Require:** at time $t$ drone $i$ is at position $P_i^t$
**Ensure:** $\tilde{I}_{best}^t$ has highest $O$ from $P_{best}^t$
**if** $O(\tilde{I}_{best}^t) < T$
1: $L^t = \text{line}(P^t, SD, s), s \geq c_4$
2: $V_i^{t+1} = c_3 \cdot SD + c_5 \cdot (L_i^t - P_i^t)$
3: $P_i^{t+1} = P_i^t + V_i^{t+1}$
**else**
4: $\tilde{I}_{best}^t = \sum_{i,t}(I_i^t)$ from $P_{best}^t$
5: $V_i^{t+1} = c_1 \cdot \text{norm}(R) + c_2 \cdot \text{norm}(P_{best}^t - P_i^t)$
6: $P_i^{t+1} = \text{Rutherford}(P_i^t + V_i^{t+1}, c_4)$
7: update $SD, c_3$ wrt. target appearance
**end if**

It requires five hyper-parameters that are discussed in more detail in Hyper-Parameters: $c_1$ is the cognitive coefficient, which refines each particle's position randomly, $c_2$ is the social coefficient, which refines the position of the entire swarm toward the best sampling position ($P_{best}^t$, indicated in yellow in Figs. 3–5) at time instance $t$, $c_3$ is the scanning speed of the swarm if nothing is found (i.e., for the default scanning pattern), $c_4$ is the minimal sampling distance (minimal horizontal distance of drones), and $c_5$ (0..1) is the speed of particles' divergence back toward the default scanning pattern if the target is lost. Note that $SD$ is the normalized vector between the most recent target position and the swarm's center of gravity at that time, $R$ is a random vector, and that positions and velocities are in 2D (defined on the horizontal scanning plane that is parallel to the ground plane).

Lines 1–4 implement our conditioned bias toward a default scanning pattern if nothing is found or a previously found target is lost. Our default scanning pattern (defined by the function lines and stored in $L^t$) is linear (i.e., all drones in a line) and centered at the center of gravity of all drone positions ($P^t$) at time instance $t$, spaced at distance $s \geq c_4$, and moving at speed $c_3$ along the scanning direction $SD$. A linear scanning configuration moving in perpendicular direction ensures the widest ground coverage. As for single drones, scanning direction and speed can be defined by waypoints. The speed of divergence toward the default scanning pattern if a target is lost is $c_5$ (0..1).

Lines 5–10 implement the convergence of the swarm if a potential target is detected. The images captured by all drones ($i$) at time ($t$) are integrated (i.e., registered) to all ($i$) perspectives. The perspective at which the computed integral has the highest objective is the best integral image ($\tilde{I}_{best}^t$) and its corresponding reference pose is the best sampling position ($P_{best}^t$). Once ($P_{best}^t$) is determined, all single images ($I_i^t$) captured by all drones ($i$) at all sampling times ($t$) are further integrated to ($\tilde{I}_{best}^t$) (i.e., registering to $P_{best}^t$ and summing) only if integrating $I_i^t$ improves our objective (i.e., increases visibility in $\tilde{I}_{best}^t$). We then determine the velocities $V_i^{t+1}$ for the next time instance $t + 1$ by our cognitive ($c_1$) and social ($c_2$) components. These velocities are applied to determine the next sample positions $P_i^{t+1}$. The minimal distance ($c_4$) constraint is enforced by Rutherford scattering[69]. Finally, the direction ($SD$) and speed ($c_3$) of the default scanning pattern are updated with respect to the detected target appearance (i.e., its position and motion speed/direction). Position and motion parameters can be computed from two consecutive iterations where the target was detected. Here, $c_3$ is the distance by which the target moves during the iteration time (i.e., duration of the most recent iteration) plus some delta that guarantees that the swarm can keep pace with the target (i.e., the swarm is faster than the target). The updated $SD$ and $c_3$ become effective when the

target is lost. In this case, the swarm diverges at an appropriate speed toward the default scanning pattern in the direction of the most recent target appearance.

Our objective function ($O$) is explained in detail in Objective function. Its result is compared to a defined limit ($T$) for distinguishing a potential target signal from false positives. Note that we repeat our PSO iterations until, for example, the objective is high enough to clearly indicate a finding or the process is aborted manually. More details are presented in the following sections.

**Collision avoidance**. For simple collision avoidance, we operate the drones at various altitudes with uniform height differences of $\Delta h$. Thus, the maximum height difference between the highest and the lowest drone is $\Delta h_{max} = \Delta h \cdot n$ for $n$ drones in the swarm. Although different sampling altitudes lead to variations in spatial sampling resolution on the ground, this has almost no effect on our integral images.

The ground coverage of a drone depends on the field of view ($fov$) of its camera[57], and is:

$$c_l = \left(2 \cdot h_l \cdot \tan\left(\frac{fov}{2}\right)\right)^2 \tag{2}$$

for the lowest, and

$$c_h = \left(2 \cdot (h_l + \Delta h \cdot n) \cdot \tan\left(\frac{fov}{2}\right)\right)^2 \tag{3}$$

for the highest drone. The average coverage in the integral image is therefore:

$$c_{avg} = \left(2 \cdot \tan\left(\frac{fov}{2}\right)\right)^2 \cdot \\ \left(h_l^2 + (h_l \cdot \Delta h \cdot (n - 1)) + \Delta h^2 \frac{2n^2 - 3n - 1}{6}\right). \tag{4}$$

The corresponding spatial sampling loss ratio due to height differences is:

$$SL_{\Delta h} = \frac{c_{avg}}{c_l} = 1 + \left(\frac{\Delta h}{h_l}\right)^2 \cdot \\ \left(\frac{2n^2 - 3n + 1}{6}\right) + \left(\frac{\Delta h}{h_l}\right) \cdot (n - 1). \tag{5}$$

The coverage of a single pixel on the ground is (for the lowest and highest drone, respectively):

$$c_{pxl} = \frac{c_l}{px}, \tag{6}$$

$$c_{pxh} = \frac{c_h}{px}. \tag{7}$$

The spatial sampling loss ratio due to pose estimation error $e$[54] is:

$$SL_e = 1 + \frac{4e^2 + 4e \cdot \sqrt{c_{pxl}}}{c_{pxl}}. \tag{8}$$

Consequently, the total spatial sampling loss ratio is:

$$SL = SL_e \cdot SL_{\Delta h}. \tag{9}$$

To avoid that drones at higher altitudes capture images of drones at lower altitudes, the maximum height difference and the vertical spacing of the drones depend on the minimal horizontal

sampling distance ($c_4$) and the cameras' field of view (*fov*):

$$c_4 = \Delta h_{\max} \cdot \tan\left(\frac{fov}{2}\right)$$
$$= \Delta h \cdot (n-1) \cdot \tan\left(\frac{fov}{2}\right),$$

(10)

where $n$ neighboring drones are vertically separated by $\Delta h$.

Assuming realistic parameters, for example, $\Delta h = 1$ m, $n = 10$, $fov = 50°$, $h_l = 35$ m, $p_x = 512 \times 512$ pixels, $e = 0.05$ m (for RTK-based GPS precision), would make the spatial sampling resolution drop from $6 \times 6$ pixels/m$^2$ (sampling at the same altitude) to $5 \times 5$ pixels/m$^2$ (sampling at different altitudes)—both including the reduction in spatial sampling resolution due to the pose estimation error, as discussed in[54]. Here, $SL_{\Delta h} = 1.28$, $SL_e = 6.57$, $SL = 8.4$, and $c_4 = 4.19$ m.

This example illustrates that, compared to sampling at the same altitude, sampling at different altitudes has a minimal impact on the spatial sampling resolution of integral images. This is due not only to the integration itself (where multiple spatial sampling resolutions are combined in case they are sampled from different altitudes), but also to slight misregistrations (due to pose estimation errors) being much more dominant than resolution differences (compare $SL_{\Delta h}$ with $SL_e$ above).

A comparison of integral images sampled at different altitudes and at the same altitude is shown in Supplementary Note 2 (cf. Supplementary Fig. S2).

Note that the altitude differences for our default (linear) scanning pattern, as explained in Particle swarm optimization, were chosen to alternate equally over space as follows: highest, third-highest, fifth-highest, etc. altitude from the outer-most position of one side inwards, and second-highest, fourth-highest, sixth-highest, etc. altitude from the outer-most position of the opposite side inwards. This maximizes the overlapping ground coverage[57]. Higher altitudes for specific drones to avoid downwash can be incorporated without affecting the overall approach.

**Objective function**. If the swarm consists of $n$ drones that sample over $\tau$ time instances, we capture a total of $n \cdot \tau$ single images. At each time instance $t$, $n$ images are captured in parallel, while $n$-tuples of parallel samples are captured sequentially over $\tau$ steps.

To compute the integral image, we first apply a Reed–Xiaoli anomaly detector[70] to the $n$ latest single images captured at the current time instance $t$ to detect pixels that are abnormal with respect to the background statistics. Note that all images captured contain four spectral bands (cf. Fig. 3b): red, green, blue, far infrared/thermal. Consequently, we detect anomalies in color and temperature, which has proven to be more efficient than color anomaly detection or thermal anomaly detection alone[71].

We then integrate the masked results (i.e., binarized after anomaly score thresholding) to cope with different amounts of image overlap during swarm sampling, using a constant anomaly score threshold (i.e., $0.9998 = 99.98\%$ in all our experiments). The registration of these images is relative to one of the $n$ perspectives. Without occlusion, the choice of the reference perspective is irrelevant, as the target's footprint would only shift to different pixel positions in the integral image for each reference perspective, but would not change otherwise. With occlusion, however, we can gain more visibility from some reference perspectives than from others due to varying visibility from different perspectives. Therefore, we determine the best integral $\tilde{I}_{\text{best}}^t$ with the highest $O(\cdot)$ and its corresponding reference pose $P_{\text{best}}^t$.

Once $P_{\text{best}}^t$ is known, all single images captured at previous time steps are also anomaly masked and integrated to $\tilde{I}_{\text{best}}^t$ for $P_{\text{best}}^t$ – but only if they improve the objective (i.e., if they enhance

visibility and in the case of *fov* overlap—which indicates overlapping coverage on the ground). Finally, we remove previously integrated single images of the $n$ latest set from $\tilde{I}_{\text{best}}^t$ if this also leads to an improvement of our objective.

Note that $\tilde{I}_{\text{best}}^t$, $P_{\text{best}}^t$, and objective ($\tilde{I}_{\text{best}}^t$) are required for our PSO interaction (algorithm 1, lines 1, 6, and 7).

Our hypothesis is that the visibility of the target improves with more integrated samples[54]. In the absence of occlusion, for instance, the target should appear fully visible in the integral image, and its projected pixel footprint should have the maximum size from oblique viewing angles. However, this footprint is reduced with increasing occlusion, as only fractions of it might be reconstructable (i.e., parts that are fully occluded in all or most samples will remain invisible). It is also reduced by less oblique viewing angles.

Therefore, our $O(\tilde{I}_{\text{best}}^t)$ function determines the contour size of the largest connected pixel cluster (i.e., blob, cf. Fig. 3b) among the abnormal pixels detected and integrated in $\tilde{I}_{\text{best}}^t$, using the raster chain tree algorithm[72]. This contour size is our objective score. According to our hypothesis, an improvement in the objective score leads to an increase in visibility, making it a dependable metric for evaluating the effectiveness of detecting occluded targets. Our previous research[61] has also indicated a strong correlation between visibility and detection rate of a classifier.

The center of gravity of the blob contour represents the estimated target position.

**Hyper-parameters**. After convergence due to a potential finding, our PSO iterations approximate a solution to the packing circles in a circle problem[73], as illustrated in Fig. 6.

Here, the inner circles represent the possible locations of each drone at time instance $t$, while the outer circle represents the SA area being sampled. To guarantee the minimal horizontal search distance, it follows that $c_1 + c_2 \leq c_4$. To avoid that the cognitive search behavior of the swarm overrules its social search behavior, it follows that $c_1 \leq c_2$.

The sampling rate in the default scanning direction is defined by $c_3$ (i.e., images in the default scanning direction are taken every $c_3$ meters). After a target has been detected, $c_3$ should be chosen to be larger than the distance the target can move during the iteration time (which can be determined automatically), as explained in Particle swarm optimization. Note that $s \geq c_4$ represents the sampling rate in the orthogonal direction. The smoothness of the divergence toward the default sampling pattern after the target has been lost is controlled by $c_5$ (0..1). The larger,

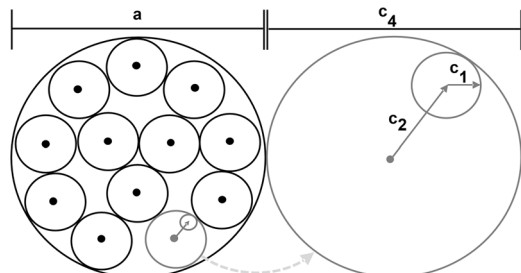

**Fig. 6 Circle packing approximation with particle swarm optimization.** Particle swarm optimization's approximation to a packing circle in a circle solution: The possible movement of each drone in one Particle swarm optimization iteration is $c_1 + c_2$, and must be less than or equal to the minimal horizontal search distance $c_4$. *a* represents the synthetic aperture area, $c_1$ and $c_2$ are the cognitive and social components, respectively.

the quicker the divergence. Low values (e.g., $0.3 = 30\%$) should be chosen for smooth transitions.

Note that the iteration time varies, as it equals the required duration of the drone with the longest travel distance to reach its position. To avoid oversampling, $c_2 - c_1$ (which is the smallest possible distance a drone can move in each iteration) must not be less than the minimum sampling baseline, which depends on projected occluder sizes, as explained in ref. [54].

Finally, the diameter $a$ of the SA is $a = c_4 \cdot r_n$, where $r_n$ is the packing number[74,75] for $n$ circles (i.e., drones) of the packing circles in a circle problem[73]. For example, the SA diameter for $n = 10$ drones at minimal horizontal sampling distance of $c_4 = 4.19$ m is 15.976 m, as $r_{10} = 3.813$.

**Implementation**. Our forest simulation was realized with a procedural tree algorithm called ProcTree and was implemented with WebGL. For all our experiments, it computed $p_x = 512 \times 512$ pixels aerial images (RGB and thermal) for drone flights over a predefined area and for defined sampling parameters (e.g., waypoints, altitudes, and camera field of view). The virtual rendering camera ($fov = 50°$ in our case) applied perspective projection and was aligned with its look-at vector parallel to the ground surface normal (i.e., pointing downwards). Procedural tree parameters, such as tree height (20–25 m), trunk length (4–8 m), trunk radius (20–50 cm), and leaf size (5–20 cm) were used to generate a representative mixture of broadleaf tree species. Finally, a seeded random generator was applied to generate a variety of trees at defined densities and degrees of similarity. Environmental properties, such as tree species, foliage and time of year, were assumed to be constant, as we were interested mainly in effects caused by changing sampling parameters. Forest density was considered sparse with 300 trees/ha, medium with 400 trees/ha, and dense with 500 trees/ha.

Simulated integral images are compared with integral images captured over real forest in Supplementary Note 1.

With our centralized implementation running on a consumer PC (i5-4570 CPU, 3.20 GHz, 24 GB RAM), we achieved an average (not performance-optimized) processing time for each PSO iteration of 96 ms for all of the $n$ latest (i.e., parallelly captured) samples, and 30 ms for each of the $n \cdot \tau$ samples captured during the $\tau$ previous (i.e., sequential) time steps. For $n = 10$ and by limiting $\tau$ to 3, for example, one PSO iteration requires $960 + 900$ ms $= 1.86$ s and processes a total of 40 images. With a partially decentralized implementation, image capturing and transmission as well as anomaly detection can be carried out in parallel on each drone. Faster GPU implementations of the anomaly detector lead to an additional speed-up. While distributing the image and telemetry data collection process using one or more drones is technically feasible, a centralized approach is deemed more practical and achievable for drone swarms in our case. Recent 5G ground stations and cloud APIs can provide the required bandwidth and enable easier access to data, making the centralized approach more viable.

## Discussion and conclusion

In all our experiments, the target was found at the correct position if it was detected. The chances for a detection, however, were generally much lower in the case of blind uniform sampling (parallel or sequential) than for adaptive sampling of a swarm (e.g., 3.8×/1.4× for blind parallel/sequential sampling, cf. Figs. 2 and 3). Swarm sampling reached a particular target visibility significantly faster than blind brute force sequential sampling (e.g., 12× to reach an MTV of at least 50%, cf. Figs. 2 and 3), while blind parallel sampling was fast but never achieved adequate target visibility.

The reason why swarms significantly outperform blind sampling strategies in terms of performance and detection rate is that sampling can be adapted autonomously to locally sparser forest regions and to larger target view obliqueness. In all our experiments, swarms preferred to converge at certain distances from the target (rather than directly above it) to maximize target view obliqueness. In open fields, this would be the only possibility to improve visibility, as it increases the projected footprint of the target. In occluding forests, however, sampling through locally sparse regions is another factor to be considered. Our PSO and objective function optimize for both (sparseness and obliqueness) while also combining sequentially and parallelly recorded samples whenever possible.

The wider SA and denser sampling of larger swarms will always lead to better visibility, larger coverage, and consequently to a higher detection probability, as shown in Fig. 4a–c, f. Stronger occlusion will generally reduce visibility, as illustrated by Fig. 4c–e, g.

For static targets, visibility increases with the number ($N$) of images being integrated. Under the assumption of uniformity (size and distribution of occluders), the following statistical behavior describes the visibility ($V$) of a target in an image where occlusion appears with density ($D$)[54]:

$$V = 1 - D^2 - \left( \frac{D(1 - D)}{N} \right). \tag{11}$$

Visibility improvement has upper and lower limits that depend on $D$. In the worst case, with a single image ($N = 1$): $V_{\min} = 1 - D$. In the best case, with an infinite number of images being integrated ($N = \infty$): $V_{\max} = 1 - D^2$. Note that in the case of non-uniform occlusion volumes that are uniformly sampled, the same principle applies under the assumption that $D$ is the average density over the $N$ samples. This can be observed for the blind brute force sequential sampling shown in Fig. 2d, where $N$ was sufficiently large. It reveals the same geometric distribution behavior as for uniform occlusion volumes[54]. For non-uniform occlusion volumes which are adaptively sampled, as in the case of our drone swarms and PSO, the density of each sample cannot be considered statistically equal, as it is minimized individually. For this reason, $V$ increases much quicker and settles at a much higher value than in blind sampling, as shown in Fig. 3e.

For moving targets, integrating images captured at previous time steps does not improve visibility if the projection of the target does not overlap with its projection in the most recent recordings. In this case, they are not integrated (see line 6 of algorithm 1), and only the most recent samples that are captured parallelly at the current time step can contribute to occlusion removal. Consequently, target visibility drops during motion, and increases again when the target stops, as illustrated in Fig. 5i. Although moving targets were detected and tracked reliably in our experiments, they can be lost if they remain in excessively dense regions for too long or if they move too fast (i.e., faster than the swarm can follow). However, the movement pattern of the target has no impact on the effectiveness of our approach. Blind sampling is the only feasible method in cases where no potential target signal is detected. Here, a fixed linear sampling pattern is followed to increase the possibility of detection. This is employed when a target signal has never been detected before or is lost during movement. In the latter case, the swarm returns to blind sampling in the direction of the last known target location. It should also be noted that a linear configuration orthogonal to the exploration direction gives the highest chance to (re-)detect the target. Nevertheless, if a target is never discovered due to an initial flight in the wrong direction or significant occlusion, it will inevitably be missed. Supplementary Note 7 extends the statistical visibility model for uniform occlusion volumes and static targets

(Eq. (11)[54]) to include parallel-sequential sampling in the presence of moving targets (Eq. (S15)). It also considers the contribution to visibility improvement of overlapping target projections in sequential samples, depending on target and drone speeds. As for static targets, this model is also outperformed in adaptive sampling of moving targets in non-uniform occlusion volumes due to the individual view optimization of the PSO.

Our simulation differs from the real world: acceleration and deceleration of drones, data transmission times (e.g., images and waypoints), and errors of sensors (e.g., GPS imprecision, mechanical camera stabilization and camera noise) are not considered. We encompass a flat topography without any hills. Supporting uneven topologies will be considered in future. Compared to a real forest, our procedural forest is simplified. Although this influences performance and quality, it does not affect our finding that swarms significantly outperform blind sampling in performance and detection rate under the same conditions. Improving the simulation in rendering quality or by using physics-based methods will not change this as occlusion is the primary factor. Instead, we plan to conduct experiments with physical drone swarms (capable of flying at speeds greater than 10 m/s) in real environments. Initial experiments have revealed that commercial routers or mobile 5G ground stations, as well as modern graphics processors, are fast enough for parallel RTSP streaming and GPU-accelerated decoding of ffmpeg-encoded video transmissions. This is essential for our centralized approach.

The threshold $T$ that is needed by our PSO (see line 1 of algorithm 1) for outlier removal was always set to be slightly higher than the largest false-positive blob detected when the target is not in view. To determine it, several representative sample sets without target were considered. Automatic and adaptive determination of this threshold will be part of future work. While our results show that PSO was a suitable initial choice for addressing our problem, we plan to investigate variations of PSO and other bird swarm-inspired techniques in the future. We have demonstrated that adaptive swarms are more effective than blind sampling even with simple approaches such as PSO; however, we anticipate that employing more sophisticated swarm approaches would lead to even superior outcomes.

Our collision avoidance strategy is simple but effective for AOS. It requires neither a computational nor a communication effort. Alternatives that have the potential to reduce the minimal sampling distance $c_4$ further need to be investigated. A smaller $c_4$ leads to shorter flight distances of individual drones during each PSO iteration and, consequently, to faster reaction times of the whole swarm. Wider SAs and denser sampling can always be achieved with larger swarms.

Finally, we believe that ongoing and rapid technological development will make large drone swarms feasible, affordable, and effective in the near future—not only for military but, in particular, also for civil applications, such as search and rescue. For other SA imaging applications that go beyond occlusion removal, drone swarms have the potential to become an ideal tool for realizing dynamic sampling of adaptive wide-aperture lens optics in remote sensing scenarios.

## Data availability

All experimental data presented in this article are available at https://doi.org/10.5281/zenodo.7936352. Supplementary Data 1 is also included to provide additional supporting information.

## Code availability

The simulation code used to compute all results presented in this article is available at https://github.com/JKU-ICG/AOS (AOS for Drone Swarms).

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

## Acknowledgements

This research was funded by the Austrian Science Fund (FWF) and German Research Foundation (DFG) under grant numbers P32185-NBL and I 6046-N, as well as by the State of Upper Austria and the Austrian Federal Ministry of Education, Science and Research via the LIT-Linz Institute of Technology under grant number LIT2019-8-SEE114.

## Author contributions

O.B. developed the concept, conceived and designed the algorithm and experiments. R.J.A.A.N. and I.K. implemented the algorithm and performed the experiments. R.J.A.A.N., I.K. and O.B. analyzed the data, contributed materials, and wrote the paper.

## Competing interests

The authors declare no competing interests.
