## [Peer Review File · Communications Engineering]

Reviewers' comments:

Reviewer #1 (Remarks to the Author):

The manuscript sets to demonstrate the effectiveness of using a coordinated swarm of autonomous drones to detect and track a target (person) in a densely forested area.

The work build on the authors previous research on using synthetic aperture to combine multiple images in a single integral image where the target is more clearly visible. This strategy not only helps to highlight the target with respect to the background, but is very helpful to deal with occlusions that are caused by the trees.

In previous works, the authors have already demonstrated that the integral images strategy can be used successfully with a single drone to remove partial occlusions caused by the vegetation [26,27].

This manuscript extends the previous research by considering images collected in parallel by a swarm of drones. There are several advantages to such a parallel solution:

- 1) using multiple drones to collect images in parallel can speed up significantly the construction of an integral image that achieves good visibility of the target.
- 2) in presence of a moving target, integrating images collected sequentially (from a single drone) may be too slow and it can lead to losing the target. With multiple drones flying over the search area in parallel it is easier to maintain visibility of the target.

The main contribution of this article concerns specifically the motion strategy that is used to sample images from the area of interest. The claim from the authors is that a purely blind sampling strategy where the drones (or drone) follow some predefined trajectory (e.g. given as waypoints) is not optimal, because it does not account neither for the unknown and varying density of the vegetation nor for possible movements of the target. To overcome this limitation, the manuscript proposes a sampling strategy applicable to the multi drone setup and based on particle swarm optimization (PSO).

This particle optimization strategy aims to maximize an objective function that gives a measurement of the visibility of the target in the integral image, which is a non-linear and non differentiable function.

The problem addressed in the paper is interesting and having an effective multi-drone solution for searching and tracking people in a forest may be very important for search and rescue missions, so this research has the potential fo be very impactful. Yet, I think that the manuscript has a few weaknesses that should be addressed.

1) OBJECTIVE FUNCTION: the sampling strategy proposed in the manuscript is based on the goal of optimizing an objective function that is defined as the contour size of the largest blob of connected pixels found from an anomaly detector applied to an integral image. More specifically, the objective function is expressed as a percentage, with respect to the maximum contour detected from a target in an unoccluded scenario.

Certainly improving the visible area of the target is essential to detect it, however the goal of the task is

not to see the full target but just to detect it. Thus, it is questionable whether it is necessary to achieve a visibility score of 72% or whether a score of 50% is enough. In other words, the visibility score does not directly translate to a detection score.

I do not think that the visibility score is a proxy metric sufficient to gauge the impact of this sampling strategy in detecting the target, because it does not provide an insight to when the detector can confidently claim that the target has been found. For example, a visibility of 50% may be enough to detect with certainty a person.

Additionally, there are other metrics that would be important to consider, at least in the discussion of the results, such as the false positive rate (when anomalies that are not the target are detected and tracked by the PSO).

2) COMPARISON WITH SINGLE DRONE AND FIXED ARRAY OF CAMERAS: the experiments presented in the paper compare the drone swarm solution to scanning strategy using a single drone or a fixed array of cameras connected to a single drone. It is clear that the multi drone solution can achieve faster convergence to a good visibility configuration, considering the larger number of degrees of freedom. However, for me it is unclear why the single drone and the array of camera strategies achieve worse visibility scores. In theory, these strategies could still sample finely the research area and store all the images captured, to then integrate the best images as done in the PSO solution. Certainly this would be orders of magnitude slower, but it should be able to achieve the same scores. Additionally, the simple blind sampling for the single drone and camera array could also be improved, using an adaptive solution based on the sequence of stored images.

Thus, the presented comparison does not seem to be fair.

3) CENTRALIZED ALGORITHM: the algorithm used in the experiments is implemented in a centralized setting, with all the images being elaborated on a desktop computer. While this can be considered as a first proof of concept, in practice I don't see this as a reliable or practical solution, as it would mandate setting up a base station with enough bandwidth to communicate rapidly with all the drones. Additionally, the PSO strategy requires that all the drones share their images in order to compute the update for the particles, but there is no analysis of the cost due to the latency of the transmission, which I believe is not simulated.

I believe that a decentralized algorithm would be more useful for practical applications and it would circumvent the single point of failure of the centralized algorithm.

4) METHOD: concerning the method, I did not quite understand how many images are used to compute the integral image, nor how the best integral image is selected. Is it done performing a brute force search over the current and past collected images?

5) SIMULATIONS: unfortunately the method is only demonstrated in simulation. The supplementary does show a single comparison between a synthetic and a real integral image, but I do not deem that sufficient to prove that the method works well in practice. For instance, I can expect that in the real world the method can be a lot more sensitive to the choice of the threshold T , and perhaps the

threshold may have to be increased to exclude false positives, thus slowing down convergence. Moreover, the real world environment may present more significant domain shifts and challenges (e.g. due to change in illuminations, reflections in the NIR band) that cannot be replicated in the simulation. Considering that previous articles from the authors [26,27] contained real world experiments, I think that it would be necessary to add them here too, perhaps with a small flock of drones (e.g. 3).

6) PSO STRATEGY AND TARGET TRACKING: the PSO method proposed in the paper implements a simple binary strategy that lets the drones coordinately move in a certain direction (where the target was) to gain visibility of the target when it is lost, and then a randomized loitering strategy around the target once the objective function is above a fixed threshold (meaning that there is a good visibility of the target). However, this simple strategy has a few pitfalls:

- what happens when the target has to be yet discovered? Does it use a blind strategy, following a predefined scanning pattern? This is not explained and indeed the experiments always start conveniently with the swarm moving towards the target or with the target already in view.
- the linear exploration phase (when the objective function is below the threshold) is quite simple, and it may fail if the target moves quickly and with jerky patterns. The proposed experiments unfortunately consider only rather ideal cases, where the target only moves in a line or in a circle.
- even the authors state that "Although moving targets were detected and tracked reliably in our experiments, they can be lost if they remain in excessively dense regions for too long or if they move too fast (i.e., faster than the swarm can follow)". However, the manuscript does not present any example of such failure cases.

7) DRONE SPEED: the simulations considers the drones moving at 10 m/s. However, the previous articles [26,27], which include real world experiments, state that the flying speed is chosen below 1m/s. Thus, it is questionable whether the findings in the simulations can be replicated in the real world, and whether this solution is able to track a person moving even at a walking/jogging pace (4-5m/s).

8) MANUSCRIPT: At a first read, I found difficult to understand the Results section since it is presented before the Method. I think it would be advisable to swap the order of the two sections.

I also found a little difficult to understand the figures from the simulations, particularly Fig. 5, with all the arrows and boxes.

I believe that these weaknesses should be addressed to make the contribution of the paper more significant.

I would also suggest to look in the literature of multi aerial tracking, and particularly the work of Aamir Ahmad. Although his work is not focused on dealing with vegetation occlusions, it is relevant for this field of research.

- Deep Neural Network-Based Cooperative Visual Tracking Through Multiple Micro Aerial Vehicles, Price et al., Robotics and Automation Letters, 2018

- Active Perception Based Formation Control for Multiple Aerial Vehicles, Tallamraju et al., IEEE Robotics and Automation Letters, 2019

Reviewer #2 (Remarks to the Author):

The paper presents a method where a swarm of drones with cameras try to find and track the most optimal unconcluded view of a target person in a forest. The method is based on particle swarm optimization, and with a simulator that can generate integral images of the target in a forest, the authors showed that their method performs better than simple scanning heuristics without optimization.

Overall, the paper is well-written, and I can hardly find any grammar or spelling mistakes throughout. Additionally, the figures are quite clear and well-made. I found it challenging to follow the relationship between the methods and the discussions/conclusion. Some details that were discussed in the conclusion should have been included in the methods section for clarity.

Although the simulation showed the effect of 3 different swarm sizes and 3 different forest densities, only one simulation was presented for each scenario, which may limit the conclusions that can be drawn from the results. It would be helpful to include a table that displays all possible combinations of the simulation parameters with multiple repeats of each simulation and the average final visibility percentage and time to reach the plateau to provide a more comprehensive analysis of the results. Then, statistics could be used to determine how the quality of the search increases and decreases with each iteration.

To better understand the potential of the method, it would be useful to investigate the effect of increasing the parameters, such as the number of drones and forest density, to determine the point of diminishing returns. At one point, adding more drones will not increase the quality of the target visibility more, but now there is such a big increase in quality from 5 to 10 that it's not clear where that limit is. I'm also missing a comparison with another optimization technique. I feel that the simulation could have been used more extensively to generate even more results to get a full picture.

Regarding the discussion, as I mentioned before, many elements need to be explained in the discussion that should have been mentioned in the methods or implementation. This makes the discussion difficult to read. "Although the authors plan to test the method on real drones, it would be valuable to include a proof of concept using a physics-based simulator with high-quality rendering and realistic sensor models to demonstrate the feasibility of the method before testing on real drones.

In conclusion, I think this is interesting and relevant work, and the paper, figures, and videos are well-written/made. But I'm not sure if the results are showing the full potential of the method on a swarm of drones following a target or compared to other explorative methods, or that this has the potential to work on a real swarm of drones. I encourage the authors to make full use of the faster non physics simulator and simulate more runs in a way that it is sufficient to draw conclusions.

Detailed comments

Abstract:

- A line should be added to explain the experiments as being done in simulation

Introduction:

- line 18-24: You could also include cinematography or agriculture?
- line 118: Have you considered anything else than PSO? Stochastic Optimal control? Ant colony optimization, or other consensus techniques for swarm agent control? I'm missing a bit of the motivation of using PSO over other techniques that are suitable for non-linear systems too.

Results:

I'll be mostly giving comments here per figure.

- Fig 2 d) it is difficult to really compare parallelly sampling camera array and sequentially sampling single camera drone, as the second is much shorter. Also, it seems that the single drone was allowed to cover a much bigger area so the comparison is not on the same level
- Fig. 3 c) If the drone is able to see the center of the target through one line, eventhough perhaps a part might be still occluded in the field of view , would it still be counted as a full view for that one drone? Or how would you calculate the part that is not seen? And it also seems that the rays hit the target a too perfect... are they aware already of the exact location?
- Fig 3 e) Same thing here what I mentioned in the last figure's comment, the comparison for especially the parallel sampling camera array does not seem on the same level. It should perhaps be allowed to cover a larger area.
- Fig 4 f) It seems that (a) and (b) still have somewhat of an upward trend. It would be better to let the simulation through until the same time of seconds as (c)
- Fig 4 g) Same thing here, the lines still show an upward trend, and it would be better to continue the simulation for longer until a clear plateau is reached for all of them. Also, it would be interesting to see if the improvements of the 500 threes/ha would be improved with more swarms.
- Fig 5 i) The figure letters in the plot makes things very clear of what to look for. I'm not sure why the apart between f) and g) would consider convergence/divergence as it seems that the swarm mostly lost the biggest part of the target. Also I would have expected the swarm to be able to follow the target more, especially it being a low density forest.

Methods:

Particle Swarm optimization

Equation 1) What is V ? are those the velocity vectors? Perhaps that should be also added in the text below.

Algorithm 1: I have to search a bit far in the text what R , SD and other parameters mean, so perhaps it would be good to put it closer to the first mention of algorithm, or at least the first paragraph afterwards.

Collision Avoidance

- It's quite simple to keep this only based on height separation. If you would have 10 drones flying in a range for 10 meters, it needs to be made sure that they don't fly over each other for downwash and I sometimes do see that in simulation (although I don't know how far they are from each other).

Implementation:

- Have you considered simulating hills? Many forest are not in completely flat lands.
- Is are anything done in terms of simulating the model of the camera? Later in discussion it is mentioned that it isn't but that is something that implementation should have mentioned already.

Discussions:

- line 675 The information of the PC should probably in 'Implementation'
- line 690-701: You could that the simulation doesn't include the movement of the cameras are influenced by the motion of the drones, although this can usually be solved by a good gimbal.
- line 690-701: another step before trying this out on real drones, is to record real integral images over a large area at different altitudes and use that instead of the simulated forest as a proof of concept. Also a more physics based simulation with modelled delays and sensor noise would help out to tweak the technique further before applying it on real drones.

Supplementary videos.

Videos in general:

- The videos start from number 2 to 6 which is probably to make it complementary with the figure numbers but it is confusing. Just let it start from number 1.
- The plots does not seem to be in the same scale or units. as in the paper

Video abstract:

- Music is a bit too loud compared to the voice-over.
- I'm not a fan of autogenerated voice for the narration.

Video supplement 2

- Perhaps good to indicate where the target is at the beginning of the video?
- The plot in the video only says metric... and also, I see points that I don't see in Figure 2d that are perhaps outliers?
- It would be better to have the position of the target shown at the beginning of the video

Video supplement 3

- The time scale of the plot is too long, make it shorter to be closer to what you see in the paper?
- It would be better to have the position of the target shown at the beginning of the video

Video supplement 4

- Bit easy for the swarm? The target is right in front of their pathway
- Stopping too soon before convergence or the plateau was reached
- It would be better to have the position of the target shown at the beginning of the video

Video supplement 5

- The swarm has clear difficult of following the walking target, which I think is a bit unlikely
- the fact that I'm able to see the targets here makes things a lot clearer

Supplementary materials

S1: I appreciate the comparison of the real and simulated integral images but the amount of examples are limited, so perhaps some more examples can be shown.

S2: It's good that this is shown, but I find it unlikely that this would also be the case with real images as well. For the simulation, is the focal length and distortion of the lens also simulated?

Reviewer #3 (Remarks to the Author):

The idea of the paper is interesting, and in general, the manuscript is well-written and organized.

However, some major concerns must be addressed:

- 1 - Improve the abstract to summarize the differences in the new approach. Besides, provide quantitative results obtained from the technique.
- 2 - Figures are shown before they are called in the text and explained. Please, correct it. Otherwise, it isn't very clear, especially in section discussion and conclusions.
- 3 - The introduction must have a comparison with more similar works in the literature. A table would be interesting to summarize them.
- 4 - The section Results before the section Methods is too confusing. The information of this section (from line 185 ahead) must be added in the section Methods as an initial subsection, the other info can be removed because there isn't new information.
- 5 - Implementation and discussion must be in a section where the simulations are presented along with the quantitative and quantitative results. As the results are in the figures located at the beginning of the paper, when the section that explains them starts, it is confusing to back and check them.
- 6 - The graph containing the results is small and difficult to follow. I strongly believe that the graphs with the convergence\divergence must be split into other figures to improve quality. Besides, more results must be presented along with simulation ones. Otherwise, it is not convincing enough. Please, fix it.

We would like to thank the editor and reviewers for their valuable comments. They have been addressed as follows:

Reviewer 1

- 1) **The sampling strategy proposed in the manuscript aims to optimize an objective function based on the contour size of the largest blob of connected pixels from an anomaly detector, but the visibility score may not be a sufficient metric for gauging the impact of the strategy on detecting the target, as a visibility score does not necessarily translate to a detection score.**

As demonstrated in our previous work [85], there is a strong correlation between visibility and the detection rate of a classifier. Confidence scores are also closely linked to visibility, as classifiers can accurately distinguish between true and false positives based on features like shape. The focus of this paper is not on classification but on enhancing visibility using an adaptive swarm sampling method. Although various detection techniques exist, including classification and anomaly detection, all can benefit from improved visibility.

We have now added the following in the objective function section:

“According to our hypothesis, an improvement in the objective score leads to an increase in visibility, making it a dependable metric for evaluating the effectiveness of detecting occluded targets. Our previous research [85] has also indicated a strong correlation between visibility and detection rate of a classifier.”

- 2) **The paper presents experiments comparing a drone swarm solution to a single drone or fixed array of cameras strategy, where the multi-drone solution achieves faster convergence to a good visibility configuration, but it is unclear why the single drone and camera array strategies achieve worse visibility scores.**

Performing brute force sampling with an infinite number of samples over the entire search area would lead to the same visibility results, but it is not a viable option for time-critical applications such as search and rescue, where time and drone battery life are limited. Therefore, the proposed adaptive swarm sampling approach is compared against blind sampling (sequential and parallel) approaches in terms of visibility (y-axis of plots) and time (x-axis of plots), demonstrating superior performance. This comparison is deemed fair since the objective is to find the target quickly and reliably, which is an optimization problem that cannot be solved by dense brute force search due to performance and complexity constraints.

We have already discussed in the introduction section, that our experiments show adaptive drone swarm sampling approach increasing the visibility with reduced sampling time in contrast to the blind sampling strategies (sequential and parallel). We have now added the following to the introduction section:

“We Acknowledge that blind brute force sampling with an infinite number of samples over the entire search area may yield similar visibility results, but is unsuitable for time-sensitive applications such as search and rescue operations where time and drone battery life are limited. To address this limitation, the proposed adaptive swarm sampling approach is evaluated against blind sampling methods (sequential and parallel) based on visibility (%) and time (seconds), demonstrating superior performance. This comparison is considered equitable because the primary goal is to locate the target swiftly and reliably, which is an optimization issue that cannot be resolved by dense brute force search due to performance and complexity restrictions.”

- 3) **The centralized algorithm used in the experiments is considered a first proof of concept, but I believe that a decentralized algorithm could be more practical and reliable for practical applications due to the potential issues with bandwidth and latency.**

While it's possible to decentralize the image and telemetry data collection process by having one or more drones, a centralized approach for drone swarms is more practical and feasible due to the availability of new 5G ground stations and cloud APIs that can provide the required bandwidth and easy access to data. As a result, a centralized approach will be adopted in the physical implementation of the project over the next years.

We have now added the following in the Implementation section:

“While distributing the image and telemetry data collection process using one or more drones is technically feasible, a centralized approach is deemed more practical and achievable for drone swarms in our case. Recent 5G ground stations and cloud APIs can provide the required bandwidth and enable easier access to data, making the centralized approach more viable.”

- 4) **It is unclear as to how many images were used to compute the integral image and how the best integral image was selected. Is it done performing a brute force search over the current and past collected images?**

We do not employ brute force search in determining the best integral image. At first, we integrate the RX masked results of all the single images captured at time ‘t’ using ‘n’ drones. The registration of these images is relative to one of the ‘n’ perspectives. The perspective at which the computed integral has the highest objective is the best integral image (\tilde{I}_{best}^t) and its corresponding pose is the best sampling position (P_{best}^t). The best integral image will initially contain ‘n’ images. However, once (P_{best}^t) is known all the single images captured from previous time steps are also anomaly masked and integrated to (\tilde{I}_{best}^t) for (P_{best}^t) but only if they improve the objective (i.e., they enhance visibility.)

To clarify this, we now add the following to the particle swarm optimization section:

“Lines 5-10 implement the convergence of the swarm if a potential target is detected. The images captured by all drones (i) at time (t) are integrated (i.e., registered) to all (i) perspectives. The perspective at which the computed integral has the highest objective is the best integral image (\tilde{I}_{best}^t) and its corresponding reference pose is the best sampling position (P_{best}^t). Once (P_{best}^t) is determined, all single images (I_i^t) captured by all drones (i) at all sampling times (t) are further integrated to (\tilde{I}_{best}^t) (i.e., registering to (P_{best}^t) and summing) only if integrating (I_i^t) improves our objective (i.e., increases visibility in (\tilde{I}_{best}^t)).”

- 5) **The method has been demonstrated in simulation. I suggest that real-world experiments should also be conducted to account for the challenges presented in a real-world environment.**

In the article, we have discussed the limitations of the simulation and mentioned that we plan to experiment with physical drone swarms in real environments in the future. Currently, the physical implementation of the proposed approach has been granted funding and the corresponding research project is scheduled to start in the upcoming spring. The process of implementing and evaluating the physical system is expected to take a minimum of three years. However, the challenges associated with real-world measurements, such as sensor noise, will affect both drone swarms and individual drones that sample blindly. If our simulation results indicate that adaptive drone swarms are superior to blindly sampling drones (as supported by our field experiment results), we expect to see similar

outcomes in physical experiments, even though the overall results for both cases will be inferior to those obtained from the simulation. And we already have presented many physical results for blind sampling.

We have discussed the following in discussion and conclusion section:

“Our simulation differs from the real world: Acceleration and deceleration of drones, data transmission times (e.g., images and waypoints), and errors of sensors (e.g., GPS imprecision and camera noise) are not considered. Compared to a real forest, our procedural forest is simplified. Although this influences performance and quality, it does not affect our finding that swarms significantly outperform blind sampling in performance and detection rate under same conditions.”

We have now added the following in discussion and conclusion section:

“Instead, we plan to conduct experiment with physical drone swarms (capable of flying at speeds greater than 10m/s) in real environments.”

6) The PSO method proposed in the paper has a simple binary strategy, but it lacks an explanation of how it will discover a target if it hasn't been detected before.

In cases where no potential target signal is detected, blind sampling becomes the only viable option. The swarm adheres to a fixed linear sampling pattern to maximize the likelihood of detecting a target signal. This method is employed when a target signal has never been detected before or if the target signal is lost during motion. In the latter case, the swarm diverges back to blind sampling towards the direction of the last known location of the target. However, if a target signal is never detected due to reasons such as initially flying in the wrong direction or significant occlusion, then the target will inevitably be missed.

We have now added the following to the discussion and conclusion section:

“Blind sampling is the only feasible method in cases where no potential target signal is detected. Here, a fixed linear sampling pattern is followed to increase the possibility of detection. This is employed when a target signal has never been detected before or is lost during movement. In the latter case, the swarm returns to blind sampling in the direction of the last known target location. It should also be noted, that a linear configuration orthogonal to the exploration direction gives the highest chance to (re-)detect the target. Nevertheless, if a target is never discovered due to an initial flight in the wrong direction or significant occlusion, it will be missed inevitably.”

- The simple linear exploration phase of the proposed PSO method may fail if the target moves quickly with jerky patterns.

As discussed in the paper, as long as the target is visible to the swarm, the motion being smooth or jerky would not be an issue, except when the target moves too fast or gets entirely occluded for a prolonged period, leading to the loss of the target.

We have discussed the following in discussion and conclusion section:

“Although moving targets were detected and tracked reliably in our experiments, they can be lost if they remain in excessively dense regions for too long or if they move too fast (i.e., faster than the swarm can follow).”

We have now added the following in discussion and conclusion section:

“However, the movement pattern of the target has no impact on the effectiveness of our approach.”

“It should also be noted, that a linear configuration orthogonal to the exploration direction gives the highest chance to (re-)detect the target.”

- **The authors acknowledge that moving targets may be lost if they stay in dense regions for too long or move too fast, but the paper does not provide an example of such case.**

We have now provided two examples of failed cases that resulted from the target moving too fast and in too dense regions in the supplementary section S4 and S5 respectively.

- 7) **The simulations in the paper consider drones moving at 10 m/s, but previous real-world experiments [26,27] with drones suggest that the flying speed is usually below 1 m/s, raising doubts about the replicability of the simulation's findings and the solution's ability to track a person moving at a walking/jogging pace.**

The papers mentioned are outdated wrt. speed and were based on AOS prototypes that could not fly faster than 1m/s. However, the drones used for AOS today are capable of flying at a speed of 10m/s or more (currently up to 19m/s). More information can be found in [81] and at the GitHub page for AOS on DJI (https://github.com/JKU-ICG/AOS/tree/stable_release/AOS%20for%20DJI).

We have now added the following to the discussion and conclusion section:

“Instead, we plan to conduct experiment with physical-drone swarms (capable of flying at speeds greater than 10m/s) in real environments.”

- 8) **I had difficulty understanding the Results section presented before the Method and therefore suggest swapping the order of the two sections. Additionally, I find it difficult to understand the figures from the simulations, especially Fig. 5, due to the abundance of arrows and boxes.**

In accordance with the standard policy of all Nature journals and also Communications Engineering, it is customary to present the results section before the methods section. As already discussed, and agreed with the editor, we need to maintain the current structure (i.e., results before methods). However, the figures from the simulations, including Fig. 5, are accompanied by captions that explain the various elements, providing additional information and context to aid in their interpretation. We also have made modifications to enhance the legibility of the figure 5(i).

- I suggest to look in particular at Aamir Ahmad's work on multi-aerial tracking, which is relevant to the field of research despite not being focused on vegetation occlusions.

Deep Neural Network-Based Cooperative Visual Tracking Through Multiple Micro Aerial Vehicles, Price et al., Robotics and Automation Letters, 2018.

Active Perception Based Formation Control for Multiple Aerial Vehicles, Tallamraju et al., IEEE Robotics and Automation Letters, 2019.

The suggested literature is added to our related work.

We now add the following to our list of references and related work discussion:

- 19). Price, E., Lawless, G., Bühlhoff, H., Black, M. & Ahmad, A. Deep neural network-based cooperative

visual tracking through multiple micro aerial vehicles. IEEE Robotics and Automation Letters PP (2018). <https://doi.org/10.1109/LRA.2018.2850224>.

20). Tallamraju, R. et al. Active perception-based formation control for multiple aerial vehicles. IEEE Robotics and Automation Letters PP, 1-1 (2019). <https://doi.org/10.1109/LRA.2019.2932570>.

We have now cited the suggested work in the introduction section:

*“They have been used for surveillance and environment mapping [6-13], airbase communication networking [14-18], **target detection and tracking [19-22]**, infrastructure inspection and construction [23-25], or load transport and delivery [26], and are particularly useful in areas that are not easily accessible by humans, such as forests and disaster sites [27-29].”*

Reviewer 2

The simulations examined different swarm sizes and forest densities. A comprehensive analysis could be achieved by presenting a table with all possible combinations of parameters and multiple repeats with the average final visibility percentage and time.

Our experiments were designed with a fixed set of parameters (density and number of drones) in order to study the impact of a single parameter (density or occlusion) independently of the other. It is evident that increased density invariably reduces visibility regardless of the number of drones, while increasing the number of drones yields better results due to the increased number of samples collected over time. Combining the parameters would not lead to any new findings.

We have discussed the following in the discussion and conclusion section:

“The wider SA and denser sampling of larger swarms will always lead to better visibility, larger coverage, and consequently to a higher detection probability, as shown in Fig. 4 (a-c, f). Stronger occlusion will generally reduce visibility, as illustrated by Fig. 4 (c-e, g).”

The method's potential can be better understood by investigating the impact of increasing the parameters, such as the number of drones and forest density, to determine the point of diminishing returns.

In the paper, we have established that increasing the number of drones improves the results by providing more samples for a larger synthetic aperture and therefore better visibility in a shorter time. However, if the number of drones is too high or the synthetic aperture is too large, then the field of view coverage of the drones (cameras' overlapping ground surface region) could become zero. Furthermore, the density versus visibility parameter has already been extensively evaluated in [78].

We have now added the following to the results section:

*“The size of the swarm clearly matters, as shown in the experiment in Fig. 4 (a-c, f). Larger swarm's profit from a wider SA and a denser sampling. They consequently led to better visibility (max. 72% for n=10, 35% for n=5, and 22% for n=3) and larger sampling coverage. **However, if the swarm becomes too large or the SA is extremely wide, then the field of view coverage of the drones' cameras (i.e., the overlapping ground region covered by the swarm) reduced to zero for drones at the SA's periphery.**”*

I'm also missing a comparison with another optimization technique.

We acknowledge that PSO was a suitable initial choice for our problem and our results demonstrate that adaptive swarms outperform blind sampling. We anticipate that employing superior swarm approaches would yield even better results and therefore intend to explore variations of PSO and alternative bird swarm inspired techniques in the future.

We have now added the following in the Discussion and Conclusion section:

“While our results show that PSO was a suitable initial choice for addressing our problem, we plan to investigate variations of PSO and other bird swarm inspired techniques in the future. We have demonstrated that adaptive swarms are more effective than blind sampling even with simple approaches such as PSO; however, we anticipate that employing more sophisticated swarm approaches would lead to even more superior outcomes.”

A proof of concept using a physics-based simulator is recommended before testing on real drones.

We have applied procedural forest simulation in other AOS investigations before, compared it with physical results of field studies. Based on our experience with AOS and this simulation over the years, we are confident that our procedural forest model and sensor simulations are adequate to demonstrate the superiority of adaptive swarm sampling over blind sampling. Occlusion, which is adequately simulated, is the primary factor in this. Improving rendering quality or physics-based simulation would have no effect on this since it would impact both adaptive and blind sampling methods equally. While we are working on implementing the technique on actual drones, it will not be completed or evaluated for at least three years. We recognize that our results are presented using simulations and that a physical implementation with real drones is on the way (but not available within the next three years). The latter is, in our opinion, not necessary to make the point of our current work presented in this paper.

We have discussed the following in the Discussion and Conclusion section:

“Our simulation differs from the real world: Acceleration and deceleration of drones, data transmission times (e.g., images and waypoints), and errors of sensors (e.g., GPS imprecision and camera noise) are not considered. Compared to a real forest, our procedural forest is simplified. Although this influences performance and quality, it does not affect our finding that swarms significantly outperform blind sampling in performance and detection rate under same conditions.”

We have now added the following in the Discussion and Conclusion section:

“Improving the simulation in rendering quality or by using physics-based methods will not change this as occlusion is the primary factor. Instead, we plan to conduct experiment with physical-drone swarms (capable of flying at speeds greater than 10m/s) in real environments.”

The work is interesting and well-presented, but I am unsure if the results show the method's full potential for drones and encourage to use a faster non physics simulator to run more simulations.

It should be noted that this paper does not assert that the proposed swarm approach is optimal for our problem; rather, it presents initial evidence that adaptive swarm sampling outperforms blind sampling in our particular case. This has been demonstrated, and any potential enhancements to the swarm approach or objective function will be the subject of future work.

Detailed comments

Abstract:

- A line should be added to explain the experiments as being done in simulation

We have now added the following to the abstract:

“Our simulated experiments reveal that adaptively sampling drone swarms achieve a maximum target visibility of 72% within 14 seconds. In comparison, blind sequential brute force sampling resulted in only 51% visibility after 75 seconds, while the blind parallel sampling attained 19% visibility in just 3 seconds.”

Introduction:

- line 18-24: You could also include cinematography or agriculture?

We believe that this is out of scope of our current work. We rather would like to focus on our main applications without concrete evidence and experience in other fields. But thank you for the suggestions, we'll keep them in mind.

- line 118: Have you considered anything else than PSO? Stochastic Optimal control? Ant colony optimization, or other consensus techniques for swarm agent control?

Off course we did consider other approaches before making a decision on using PSO. Many other approaches are not suitable for us (e.g., we are not optimizing paths, like possible with ant colony, but parallel sampling patterns), while having constraints in sequentially changing these sampling patterns (in straight segments), e.g. due to flight times (i.e., genetic algorithms are not applicable here). So, birds swarm inspired approaches are relevant for us. PSO might not be the optimal choice in this domain, and we will investigate better alternatives in future. However, we already show that even with PSO we can outperform blind sampling. A better swarm approach will just enhance this further. The reason for choosing PSO in general is already presented in the Particle Swarm Optimization section. As explained before, we plan to explore various PSO modifications and alternative bird swarm inspired methods in the future. However, since our study is not focused on path optimization, techniques such as Ant Colony Optimization are outside the scope of our investigation.

We have discussed the following in Methods section:

“Our objective function is based on randomness (forest occlusion) and is (especially in the case of target motion) not constant. It is neither linear nor smooth, and certainly not differentiable. The latter is the reason for choosing PSO in general, as gradient-decent-based optimizations are not possible.”

As discussed previously, we have added the following to the discussion and conclusion section:

“While our results show that PSO was a suitable initial choice for addressing our problem, we plan to investigate variations of PSO and other bird swarm inspired techniques in the future. We have demonstrated that adaptive swarms are more effective than blind sampling even with simple approaches, such as PSO; however, we anticipate that employing more sophisticated swarm approaches would lead to even more superior outcomes.”

Results:

- Fig 2 d) it is difficult to compare parallelly sampling camera array and sequentially sampling single camera drone, as the first is much shorter. Also, it seems that the single drone was allowed to cover a much bigger area so the comparison is not on the same level.

The synthetic aperture size of the blind brute-force sequentially sampling single camera drone was 36 m and is comparable to the synthetic aperture size of the drone swarms in default scanning pattern. However, for the blind parallel sampling camera array, the synthetic aperture size was 9 m, as indicated in [87]. Larger camera arrays (and with that, larger synthetic apertures) are impracticable, as they were found to be difficult to handle and can't maintain a stable flight. Consequently, it would not be practical to employ an imaging system with a greater camera array. As a result, the blind parallel sampling camera array covers a smaller area compared to the blind brute-force sequentially sampling single camera drone. Both simulations therefore represent realistic use cases.

We have now added the following in the Results section:

"The single camera drone that carries out blind-brute force sampling has a SA size similar to that of the adaptive swarm approach in the default scanning pattern. In contrast, the blind parallel sampling camera array has a SA size of only 9 m. Larger arrays are impractical because they are difficult to maintain and do not allow proper flight stabilization due to the extreme lever arms, as explained in [87]."

- Fig 3 c) If the drone is able to see the center of the target through one line, even-though perhaps a part might be still occluded in the field of view, would it still be counted as a full view for that one drone? Or how would you calculate the part that is not seen? And it also seems that the rays hit the target a too perfect... are they aware already of the exact location?

To clarify, the rays (the lines from the drones' centers to the targets' centers in the images and videos) are utilized solely for visualization purposes, and do not have any influence on the results obtained from processing the real images captured by the drones. The degree to which the person is visible in these images varies based on the level of local occlusion, as explained with our objective function.

We have now added the following in the Fig. 3 caption:

"Note that the white rays (solely used for visualization purposes) indicate direct line of sight between drones and target."

- Fig 3 e) Same thing here what I mentioned in the last figure's comment, the comparison for especially the parallel sampling camera array does not seem on the same level. It should perhaps be allowed to cover a larger area.

As mentioned previously:

We would like to emphasize that deploying a 1D camera array for parallel sampling on a drone with a synthetic aperture size equivalent to sequentially sampling single camera drone, is not practical and in fact infeasible. The large lever arms would not allow flight stabilization anymore. The 9m was maximum for our prototype, and even with this the flight controller had to be reprogrammed to avoid swinging oscillations until crash. The motors of the (even our 5kg octocopter platform) was not able to handle any longer arrays.

We have now added the following in the Results section:

“The single camera drone that carries out blind-brute force sampling has a SA size similar to that of the adaptive swarm approach in the default scanning pattern. In contrast, the blind parallel sampling camera array has a SA size of only 9 m. Larger arrays are impractical because they are difficult to maintain and do not allow proper flight stabilization due to the extreme lever arms, as explained in [87].”

- Fig 4 f) It seems that (a) and (b) still have somewhat of an upward trend. It would be better to let the simulation through until the same time of seconds as (c)

Our simulations were longer, but for clarity and visualization reasons, the plots were truncated at the point where the objective did not improve further. Consequently, there is no better optimum in all our simulations that is not shown in the plots. We make the full simulation results available in our supplementary material.

We have mentioned the following in the figure captions:

“Note that the plots end in case of no further visibility improvement.”

We have now added the following in the Results section:

“Note, that our simulations were conducted for an extended period of time but the plots in Figs. 2- 4, were cut-off at the point where there was no further improvement observed in the objective. The full simulation results are available in the supplementary material.”

- Fig 4 g) Same thing here, the lines still show an upward trend, and it would be better to continue the simulation for longer until a clear plateau is reached for all of them. Also, it would be interesting to see if the improvements of the 500 trees/ha would be improved with more swarms.

First part: See comment above.

Second part: Our experiments were designed to examine the influence of a single parameter (density or occlusion), while maintaining a fixed set of parameters (density and number of drones). It is clear (and shown) that increased density consistently results in decreased visibility, regardless of the number of drones, and that increasing the number of drones produces superior outcomes due to the accumulation of more samples over time. As explained and addressed above: ***However, if the swarm becomes too large or the SA is extremely wide, then the field of view coverage of the drones' cameras (i.e., the overlapping ground region covered by the swarm) reduced to zero for drones at the SA's periphery.***

We also limited our simulations to a maximum of 10 drones, as more drones increases the computational complexity (procedural forest generation) of the simulations extremely (simulations with 10 drones require already 8-10 hours).

- Fig 5 i) The figure letters in the plot makes things very clear of what to look for. I'm not sure why the part between f) and g) would consider convergence/divergence as it seems that the swarm mostly lost the biggest part of the target. Also, I would have expected the swarm to be able to follow the target more, especially it being a low-density forest.

In situations, where even though the moving target is still inside the swarm's field of view (the part between f and g), the swarm would still converge and diverge depending on the local occlusion condition. If the target eventually moves outside of the swarm's view, the swarm will begin to diverge

back into the default scanning pattern towards the last known location of the target. A detailed demonstration of this scenario can be observed in Supplementary Video 5.

We have discussed the following in the Results section:

“When the target leaves the swarm's view, the swarm starts to diverge into the default scanning pattern towards the last known target position. When the target stops, the swarm mainly converges into a circular SA pattern. If the target is inside the swarm's view while moving, the swarm converges and diverges depending on the local occlusion situation.”

We have now added the following in the Results section:

“Note that its convergence and divergence behavior may also be attributed to the target moving in and out of the swarm's field of view. This is illustrated in Fig. 5(i) (f-g)”

Methods:

Particle Swarm optimization

Equation 1) What is V? are those the velocity vectors? Perhaps that should be also added in the text below.

Algorithm 1: I have to search a bit far in the text what R, SD and other parameters mean, so perhaps it would be good to put it closer to the first mention of algorithm, or at least the first paragraph afterwards.

‘V’ - refers to velocity vector. The parameter R is a random vector and SD (scanning direction) is the normalized vector between the most recent target position and the swarm's center of gravity at that time.

We have now added the following to the particle swarm optimization section:

“where V_i^t is the velocity vector of particle i at time t”

We have now added the following to the particle swarm optimization section in the first paragraph after algorithm 1:

“Note that SD is the normalized vector between the most recent target position and the swarm's center of gravity at that time, R is a random vector, and that positions and velocities are in 2D (defined on the horizontal scanning plane that is parallel to the ground plane).”

Collision Avoidance

- It's quite simple to keep this only based on height separation. If you would have 10 drones flying in a range for 10 meters, it needs to be made sure that they don't fly over each other for downwash and I sometimes do see that in simulation (although I don't know how far they are from each other).

The drones operate at predetermined altitude intervals, with a minimum altitude difference of one meter being determined by GPS precision. However, if certain drones require a higher altitude to prevent downwash, this can be taken into consideration without affecting the rest of the approach. Note also, the drones will not hover above each other, but they will bypass very quickly. We believe that downwash is not a serious problem.

We now mention the following in the collision avoidance section:

“Higher altitudes for specific drones to avoid downwash can be incorporated without affecting the overall approach.”

Implementation:

- Have you considered simulating hills? Many forests are not in completely flat lands.

The study area consists of flat terrain with no hills. However, it should be noted that AOS is capable of handling hilly terrain, unlike PSO which assumes a flat ground surface. This is a topic for future research, but will be relatively straight forward to address.

We have now added the following in the discussion and conclusion section:

“We encompass a flat topography without any hills. Supporting uneven topologies will be considered in future.”

- Is anything done in terms of simulating the model of the camera? Later in discussion it is mentioned that it isn't but that is something that implementation should have mentioned already.

We simulate both RGB and thermal cameras, including their extrinsic and intrinsic parameters, such as field of view. However, we do not account for sensor noise in the simulation, as previously discussed. Nonetheless, the effect of sensor noise on integral images is minimal. Additionally, we provide a comparison between the simulated and actual RGB and thermal images in the supplementary.

Discussions:

- line 675 The information of the PC should probably in 'Implementation'

The information regarding the PC (centralized approach) is now mentioned in the Implementation section, as explained above.

- line 690-701: You could that the simulation doesn't include the movement of the cameras are influenced by the motion of the drones, although this can usually be solved by a good gimbal.

While our actual drones have an excellent mechanical camera stabilization system, any errors that cause imprecision will affect both drone swarms and blind sampling. Therefore, we believe that comparing adaptive swarm sampling to blind brute-force sampling under the same conditions to demonstrate the former's superiority is justified. In fact, serious camera motion would be problem for our previous blind sampling AOS approaches being used with physical platforms and in field experiments. However, it is not because of a good mechanical gimbal stabilization.

We have now added the following to the Discussion and Conclusion section:

*“Our simulation differs from the real world: Acceleration and deceleration of drones, data transmission times (e.g., images and waypoints), and errors of sensors (e.g., GPS imprecision, **mechanical camera stabilization** and camera noise) are not considered.”*

- line 690-701: another step before trying this out on real drones, is to record real integral images over a large area at different altitudes and use that instead of the simulated forest as a proof of concept. Also, a more physics-based simulation with modelled delays and sensor noise would help out to tweak the technique further before applying it on real drones.

This is not viable as the sampling process depends on previous recordings itself. Achieving a uniformly high-resolution coverage of a large forest patch in an adequate resolution is unfeasible. Furthermore, this approach would also not allow for the evaluation of moving targets.

Supplementary videos.

Videos in general:

- The videos start from number 2 to 6 which is probably to make it complementary with the figure numbers but it is confusing. Just let it start from number 1.

We have implemented the suggested changes by adjusting the video numbers accordingly.

- The plots do not seem to be in the same scale or units. as in the paper

We have now made changes to ensure that the plots in the supplementary videos are consistent with those in the paper in terms of scale and units.

Video supplement 2

- Perhaps good to indicate where the target is at the beginning of the video?

We have now indicated the target's position with a yellow box at the beginning of the video.

- The plot in the video only says metric... and also, I see points that I don't see in Figure 2d that are perhaps outliers?

We have now made changes to the plot in the video to ensure consistency with Figure 2d.

- It would be better to have the position of the target shown at the beginning of the video

We have now indicated the target's position with a yellow box at the beginning of the video.

Video supplement 3

- The time scale of the plot is too long, make it shorter to be closer to what you see in the paper?

The suggested change has now been made to the supplementary video 3.

- It would be better to have the position of the target shown at the beginning of the video

We have now indicated the target's position with a yellow box at the beginning of the video.

Video supplement 4

- Bit easy for the swarm? The target is right in front of their pathway

We aim to demonstrate the convergence of the swarm. Simulating scenarios where the swarm fails to detect the target at the beginning serves no purpose as it cannot be found.

- Stopping too soon before convergence or the plateau was reached

We emphasize once again that the plots were truncated at the point where the objective ceased to exhibit improvement, leading to the absence of an upward trend. See comments above.

- It would be better to have the position of the target shown at the beginning of the video

We have now indicated the target's position with a yellow box at the beginning of the video.

Video supplement 5

- The swarm has clear difficulty of following the walking target, which I think is a bit unlikely

- the fact that I'm able to see the targets here makes things a lot clearer

As mentioned previously:

In situations, where even though the moving target is still inside the swarm's field of view, the swarm would still converge and diverge depending on the local occlusion condition. If the target eventually moves outside of the swarm's view, the swarm will begin to diverge back into the default scanning pattern towards the last known location of the target. This subsequent convergence and divergence could give an impression that the swarm has difficulty in following the target which is not the case.

Supplementary materials

S1: I appreciate the comparison of the real and simulated integral images but the number of examples is limited, so perhaps some more examples can be shown.

As previously stated, a comparison between real and simulated integral images has already been presented. Additional examples are unlikely to provide any substantial differences or new insights.

S2: It's good that this is shown, but I find it unlikely that this would also be the case with real images as well. For the simulation, is the focal length and distortion of the lens also simulated?

The simulation includes the focal length, but it does not account for lens distortion. However, real recordings are undistorted before being processed by AOS – consequently lens distortion is also not present in real images. Despite this, we believe the simulation to be accurate as real images are rectified and free from lens distortion.

We have now added the following in Supplementary section S2:

*"Note that **focal length is considered in the simulation** but sensor errors, such as GPS imprecision, **lens distortion** or camera noise, are not simulated."*

Reviewer 3

1 - Improve the abstract to summarize the differences in the new approach. Besides, provide quantitative results obtained from the technique.

We have included quantitative results obtained from our simulated experiments.

We have now added the following to the abstract:

"Our simulated experiments reveal that adaptively sampling drone swarms achieve a maximum target visibility of 72% within 14 seconds. In comparison, blind sequential brute force sampling resulted in only

51% visibility after 75 seconds, while the blind parallel sampling attained 19% visibility in just 3 seconds.”

2 - Figures are shown before they are called in the text and explained.

It is ensured now that figures are not shown prior to being referenced in the text. The figures are now rearranged accordingly to improve the clarity and organization of the manuscript.

3 - The introduction must have a comparison with more similar works in the literature.

We have now referenced more similar works in the introduction section.

We have now added the following:

*“They have been used for surveillance and environment mapping [6-13], airbase communication networking [14-18], **target detection and tracking [19-22]”***

4 - The section Results before the section Methods is too confusing. The information of this section (from line 185 ahead) must be added in the section Methods as an initial subsection, the other info can be removed because there isn't new information.

In accordance with the standard policy of all Nature journals and also Communications Engineering, it is customary to present the results section before the methods section. As already discussed, and agreed with the editor, we need to maintain the current structure (i.e., results before methods).

5 - As the results are in the figures located at the beginning of the paper, when the section that explains them starts, it is confusing to back and check them.

In order to minimize complexity, enhance clarity, and improve organization of the manuscript, the figures have been rearranged accordingly. Furthermore, the figures have been placed in close proximity to their corresponding discussions for ease of reference.

6 - Improve the presentation of the results by splitting the convergence/divergence graphs into separate figures and including additional results alongside the simulation data.

We have now made modifications to enhance the legibility of the figure 5(i). Additionally, separating the graphs into distinct figures would only complicate comprehension. In the supplementary section S4 and S5, we have included two instances of failure cases resulting from the target moving too fast and in too dense regions.

Reviewers' comments:

Reviewer #1 (Remarks to the Author):

The authors addressed all the concerns raised by the reviewers, giving for the most part convincing explanations or at least reasonable arguments.

The changes applied to the manuscript have improved the quality of the presentation, making it more clear the contribution and the methodology. I have particularly appreciated the addition of two simulations that demonstrate failure cases, which I believe is important to show the limitations of the method.

Overall, I think that the paper addresses an interesting topic and provides a preliminary result towards a relevant field application. It is a pity that the authors could not perform real-world experiment or use a more advanced simulation engine, however I will not keep this against them, as I understand their commitment to further pursue this research in the next years with new funding.

Reviewer #2 (Remarks to the Author):

I appreciate the efforts made to address my previous feedback. However, I still have concerns that remain.

Regarding the difficulty of using real drones, I understand the challenges involved as someone with experience in UAVs. My point, however, was that if a real platform is not used, the simulation should cover most scenarios and provide a comprehensive picture. If a physics simulator is not used, the current simulator should simulate as many scenarios and combinations as possible. While 8-10 hours with 10 drone simulations is intense, spending a few more days to confirm statistics would have been worthwhile. I was hoping to see more simulations added in the time between my last review and this one.

I am concerned about the time it takes for the simulations to run on a good PC and how this might impact the real-time approach of the swarm eventually. I don't believe 5G and cloud APIs will solve the problem of delays, and I am curious why cloud APIs weren't used for the simulation.

While some clarifications have been added to the paper, I was hoping for more improvements in the explanation of the paper's purpose and additional simulation results.

Regarding specific details, please note that "acknowledge" should be lowercase at line 145, and "vided" should be "video" at line 177. For Figure 4, I suggest showing the full simulation result in the paper as it is better to show the full result. The supplementary section should be more about including additional simulation results of the same scenario.

Regarding the rebuttal comment Video 4, I understand that simulating scenarios where the swarm fails to detect the target at the beginning may not serve any purpose, but it is still relevant for search and rescue scenarios (which is given as context in the introduction) where people need to be found with no prior knowledge.

For future rebuttals, I suggest avoiding phrases that could be perceived as a bit dismissive, such as "it is clear that." As reviewers, we put in a lot of effort to understand the paper, and if we don't fully comprehend the paper's goal, it suggests that improvements can be made. In this case, I had some difficulty understanding the purpose of the paper, and I was hoping for more clarification.

But, I appreciate the efforts made to address my feedback, but I still believe there is room for improvement in the paper's explanation of the purpose and additional simulation results.

Reviewer #3 (Remarks to the Author):

The authors have improved the paper and performed the modifications/suggestions. As a minor request, authors should perform an English review to check the readability and remove typos.

We would like to thank the editor and reviewers for their valuable comments. They have been addressed as follows:

Reviewer 1

No further requests.

Reviewer 2

- 1) Although the 10 drone simulations conducted in 8-10 hours were intensive, additional time devoted to confirming the statistics would have been valuable. I was anticipating additional simulations to be included in the manuscript since my last review.**

As suggested by you, we have now performed additional simulation runs and have included them in the supplementary section (**S3**), by incorporating additional plots that illustrate the outcomes of averaging three simulation runs for each of the six scenarios as depicted in Figure 4. Despite the randomness in forests and the PSO, the outcome of multiple simulation runs for the same scenario are highly consistent, as evidenced by the average plots that reinforce our earlier observations based on a single run for each scenario, further validating the statistics. Also, as suggested by you, the plots show the full simulation runs (see comment 5).

- 2) I am concerned about the time it takes for the simulations to run on a good PC and how this might impact the real-time approach of the swarm eventually.**

We would like to clarify that there might have been a misunderstanding. The computation time is solely required for image generation, such as forest simulation and realistic rendering. However, this is not applicable to physical drones as they capture images at 30fps. Furthermore, all other processing, including the objective function and PSO, operates in real-time, as outlined in the implementation section. Based on this, we are confident that a physical swarm will be sufficiently responsive, as also highlighted in point 3 (see also comment 3).

- 3) I don't believe 5G and cloud APIs will solve the problem of delays, and I am curious why cloud APIs weren't used for the simulation.**

The cloud API was not utilized in the simulator since there was no need to stream video or telemetry data from physical drones. Instead, we used simulations where this data was computed rather than recorded and streamed.

However, we have made progress in implementing an initial test for RTSP streaming of ffmpeg-encoded video streams. For the upcoming physical swarm implementation using DJI platforms, the DJI video transmission protocol (OcuSync 3), which is a closed protocol, supports 80Mb/s for video transmission between the drone and remote controller. This speed is considered real-time, with a lag of less than 150ms, which is suitable for fast drone flights controlled through the remote controller.

Using a common 1Gb/s Ethernet/WiFi router, we were able to forward the stream from the remote controller to a server, achieving 30fps for 1080p RGB video streams with a very short initial overall lag of a few milliseconds. Intel's 5G ground station offers 2x2Gb/s WiFi channels per drone. However, we

tested that it is unnecessary to use WiFi or 5G as all remote controllers can directly connect to a switch via Ethernet, giving us a range of 2.5-40Gb/s per drone when streaming from the remote controllers to the server and making us independent of a limited number of WiFi channels. All of this is considerably faster than the initial OcuSync 3 video transmission, so we do not anticipate any technical issues in creating a centralized solution that works efficiently. We have also determined that even GPU-accelerated video decoding on CUDA graphics cards in the server is swift enough. As a result, we have included some of this feasibility discussion in the discussion section.

We have now added the following in discussion section:

"Initial experiments have revealed that commercial routers or mobile 5G ground stations, as well as modern graphics processors are fast enough for parallel RTSP streaming and GPU-accelerated decoding of ffmpeg-encoded video transmissions. This is essential for our centralized approach."

- 4) Regarding specific details, please note that "acknowledge" should be lowercase at line 145, and "vided" should be "video" at line 177.**

The correction for "acknowledge" has been made. However, we believe "vided" is actually correct as it appears in the paper as "pro-vided" (=line-separated provided).

- 5) For Figure 4, I suggest showing the full simulation result in the paper as it is better to show the full result. The supplementary section should be more about including additional simulation results of the same scenario.**

In terms of the simulation length (i.e., the length of the plots in time), we have predefined a maximum time, which can be arbitrary long. However, the PSO stops when no further progress of the objective function is observed, or a threshold (e.g., X% visibility) is reached. The plots in the paper only show results until the maximum of the objective function is reached, as all remaining simulation steps are irrelevant for the PSO. Hence, we planned not to include them in the paper to maintain a better resolution of the important previous steps. However, we now have presented the fully simulated results (i.e., plots) for figure 4 in the supplementary section **(S3)**.

Moreover, as mentioned before we have performed additional simulation runs and have included them in the supplementary section **(S3)**, by incorporating additional plots that illustrate the outcomes of averaging three simulation runs for each of the six scenarios as depicted in Figure 4. Despite the randomness in forests and the PSO, the outcome of multiple simulation runs for the same scenario are highly consistent, as evidenced by the average plots that reinforce our earlier observations based on a single run for each scenario, further validating the statistics. The full data of these new simulations is also included in the supplementary material.

Additionally, we have made all our code available, which allows for the execution of additional simulations for specific configurations that were not covered in the paper.

- 6) For future rebuttals, I suggest avoiding phrases that could be perceived as a bit dismissive, such as "it is clear that." As reviewers, we put in a lot of effort to understand the paper, and if we don't fully comprehend the paper's goal, it suggests that improvements can be made.**

Thank you for your feedback. We greatly appreciate the effort put forth by the reviewers to provide constructive feedback, which helps us improve the quality of our work. It was not our intention to be dismissive in any way. We will take your suggestion into consideration and strive to use more appropriate language in the future. Thank you again for your valuable input.

Reviewer 3

The authors have implemented the modifications and improved the paper, and as a minor suggestion, an English review is recommended to enhance its readability and eliminate any typos.

We have carefully reviewed the manuscript to ensure readability and have made necessary corrections to eliminate any typos.

REVIEWERS' COMMENTS:

Reviewer #2 (Remarks to the Author):

I'll keep it short this time. I very much appreciated the rebuttal to clear the last things up and happy to see some extra results. Perhaps more could have been added but for the sake of the paper I would say this is good enough.

Just a small comment, in the caption of figure S3 it would be good to also mention that plot 1 and plot 3 has been done with 300(trees/ha), since you did say that for 2 and 4, the swarm size was 10 .

With that, I would say I have no further comments. Thanks!

We would like to thank the editor and reviewers for their valuable comments. They have been addressed as follows:

Reviewer 2

- 1) In the caption of figure S3 it would be good to also mention that plot 1 and plot 3 has been done with 300(trees/ha), since you did say that for 2 and 4, the swarm size was 10.**

As suggested by you, we have now added the requested changes to the caption of figure S3.